# A photoswitchable inhibitor of TREK channels controls pain in wild-type intact freely moving animals

Arnaud Landra-Willm[1,2,3], Ameya Karapurkar[4], Alexia Duveau[5], Anne Amandine Chassot[1,2,3], Lucille Esnault [6], Gerard Callejo[7,8], Marion Bied[1,2,3], Stephanie Häfner[1,2,3], Florian Lesage [2,3,9], Brigitte Wdziekonski[1,2,3], Anne Baron[2,3,9], Pascal Fossat [5], Laurent Marsollier [6], Xavier Gasull [7,8], Eric Boué-Grabot [5], Michael A. Kienzler [10,11] ✉ & Guillaume Sandoz [1,2,3] ✉

By endowing light control of neuronal activity, optogenetics and photopharmacology are powerful methods notably used to probe the transmission of pain signals. However, costs, animal handling and ethical issues have reduced their dissemination and routine use. Here we report LAKI (Light Activated K⁺ channel Inhibitor), a specific photoswitchable inhibitor of the pain-related two-pore-domain potassium TREK and TRESK channels. In the dark or ambient light, LAKI is inactive. However, alternating transdermal illumination at 365 nm and 480 nm reversibly blocks and unblocks TREK/TRESK current in nociceptors, enabling rapid control of pain and nociception in intact and freely moving mice and nematode. These results demonstrate, in vivo, the subcellular localization of TREK/TRESK at the nociceptor free nerve endings in which their acute inhibition is sufficient to induce pain, showing LAKI potential as a valuable tool for TREK/TRESK channel studies. More importantly, LAKI gives the ability to reversibly remote-control pain in a non-invasive and physiological manner in naive animals, which has utility in basic and translational pain research but also in in vivo analgesic drug screening and validation, without the need of genetic manipulations or viral infection.

Pain is an unpleasant sensory and emotional experience caused by noxious and/or potentially damaging stimuli[1] sensed by small-diameter primary sensory neurons called nociceptors. Because of the low spatiotemporal resolution of the existing electrical, pharmacological, and genetic tools, the in vivo dissection of pain pathways has been limited[2]. Furthermore, the stimuli used are invasive, which is inherent to the study of nociception, and the effects are variable. Optogenetic and photopharmacological tools enable the control of action potential firing, in vitro and in vivo, through either the expression of exogenous opsin-proteins[3,4] or the action of photopharmacological compounds, respectively[5,6]. Activation of these tools

allows remote control of sensory neuron excitability non-invasively by transdermal illumination[7–9] to modulate pain behavior, enabling control and test evaluations on the same animal with high spatiotemporal resolution. Although strongly facilitating the determination of cell and circuit function while reducing variability[3], these approaches are neither widely nor routinely used due to various burdensome requirements for their use. The ideal tool would specifically modulate the nociceptor's activity by controlling endogenous pain-related ion channels[10,11] without the need for (i) genetic manipulation, (ii) viral infection, (iii) surgery, and (iv) extensive animal housing. In addition, this compound (i) would be in its inactivated state in the dark, enabling

---

long-term experiments, (ii) would be ethically relevant by being non-invasive, (iii) would allow rapid, reproducible, and reversible control of pain in a physiological and pathological manner, and (iv) would be amenable to be used in different species. To obtain such a compound, we first need to identify a family of ion channels expressed in nociceptors that regulate their activity. Secondly, the photoswitch has to be designed and to be characterized for its properties and specificity. Ultimately, it needs to be tested in vivo on routinely used pain assays.

In this work, we develop a light-activated antagonist of pain-related potassium channels (LAKI). By controlling the Two-Pore-Domain Potassium ($K_{2P}$) channels TREK1, TREK2, and TRESK, LAKI endows non-invasively a reversible and reproducible light-control of nociception in different freely moving animal models with a spatiotemporal resolution, for several days, without the need for transfection, infection, genetic manipulation or surgery procedures.

## Results

### Choice of target
The two-pore-domain potassium ($K_{2P}$) channels TREK1, TREK2 and TRESK are highly expressed in nociceptors from the dorsal root ganglia (DRG) and trigeminal ganglia (TG)[12,13]. By being active at rest, TREK1, TREK2, and TRESK regulate nociceptor excitability by maintaining the membrane potential. Dysfunctional mutations inhibiting these channels in humans[14] generate hyperexcitability of TG and DRG neurons underlying allodynia and migraine[15], indicating their importance in pain induction. These properties make TREK1, TREK2 and TRESK channels suitable targets to regulate nociception.

### Design of the photoswitch
To design a photochromic ligand targeting $K_{2P}$ channels in nociceptors, we first sought to identify a pharmacological $K_{2P}$ modulator that could be modified to become light-sensitive with a minimal structural change[16]. We looked for small $K_{2P}$ antagonists that present motifs resembling azobenzene, the most commonly used molecular photoswitch[16]. We found a likely candidate in the compound ML365 (Fig. 1a), which has an aryl-benzamide moiety and was reported to inhibit the $K_{2P}$ channels TASK1 and TASK3[17]. Additionally, we showed that ML365 inhibits TREK1, TREK2, and TRESK, but not TRAAK (Supplementary Fig. 1). By substitution of the benzamide moiety with a diazene (-N=N-) to introduce an azobenzene within the structure (Supplementary Data synthesis), we obtained LAKI (light-activated $K^+$ channel inhibitor) (Fig. 1a). The initial photochemical characterization of LAKI showed that 95% of LAKI is in *trans*-state at 460 nm while irradiation at 365 nm yields a photostationary state (PSS) that is ~70% *cis*-LAKI. The thermal half-life of the *cis*-LAKI is 0.7 h at 21.5 °C and did not show photodegradation over ten cycles (Supplementary Fig. 2).

### LAKI endows light sensitivity to TREK1, TREK2 and TRESK $K_{2P}$ channels
To determine the pharmacological properties of LAKI on $K_{2P}$ channels, we tested whether it endows light sensitivity to TREK1, TREK2 and TRESK currents by expressing these channels in HEK293T cells. When we applied LAKI intracellularly, we did not observe any photomodulation of TREK and TRESK currents (Supplementary Fig. 3). Likewise, we did not observe any effect when LAKI was applied externally in the dark. However, alternating illumination at 365 and 480 nm efficiently blocked and unblocked ~80% of persistent TREK1, TREK2 and TRESK currents at −60 mV, a surrogate of the physiological membrane potential in primary sensory neurons[18] (Fig. 1b, c). LAKI photoblock is conserved among species since TRESK and TRAAK channel inhibitions are similar in mouse and human (Supplementary Fig. 4). This light-gated block and unblock happened on average in less than 170 and 310 ms, respectively (Supplementary Fig. 5). This light-dependent block seems to be specific since no or a negligible inhibition was observed for TRAAK, TASK1 and TASK3 (Fig. 1b, c and

Supplementary Fig. 6) or the distantly related potassium channel KCNQ1 (Supplementary Fig. 7).

Interestingly, LAKI is bi-stable, persisting without illumination in the higher energy *cis*-LAKI blocking state but being still available for a rapid return to inactive *trans*-LAKI upon illumination with 480 nm light (Fig. 1e). This long bi-stability relies on the effective concentration of *cis*-LAKI. Indeed, we observed a decrease of the block of the TRESK channel over time with a time constant of ~3 min when UV light intensity was decreased to mimic in vivo UV light illumination (Supplementary Fig. 8a, b). Finally, we performed an in-depth characterization, beginning with assessing the concentration dependency of the photoblock on TREK1, TREK2, and TRESK. We found an $IC_{50}$ of 2.45, 1.79, and 1.75 μM for TREK1, TREK2, and TRESK respectively (Supplementary Fig. 9). We next investigated the voltage dependency of the light-blocked current and found that the photocurrent amplitude depended on the holding membrane potential and reached almost zero at −80 mV (i.e., the expected reversal potential for potassium ions) (Fig. 1f, g), indicating that the photocurrent is exclusively linked to light-blocked potassium current inhibition.

### LAKI enables optical control of potassium currents in native nociceptors through an endogenous TREK1, TREK2 and TRESK photoblock
As observed in HEK293T cells (Fig. 1b, f), in small wild-type TG neurons, LAKI induced a fast, stable, and reversible photoblock of a constant current (Fig. 2a) that, again, was proportionally reduced when decreasing the membrane potential, reaching zero at −80 mV (Fig. 2b). This demonstrates that in a native system, LAKI specifically targets a time- and voltage-independent potassium current which represents ~26% of the neuronal leak current observed at −60 mV (Supplementary Fig. 10) and which is similar to the leak potassium current carried by $K_{2P}$ channels. As shown in Fig. 2c, genetic invalidation of *Trek1* and *Trek2* induced an ~80% decrease of the native light-blocked current (Fig. 2d). The 20% remaining current was significantly further reduced by the expression of the specific TRESK dominant negative form TRESK-MT1[15] (Fig. 2c, d). Therefore, in nociceptors, LAKI regulates the background $K^+$ current carried by endogenous TREK1, TREK2 and TRESK channels.

Together, these results show that LAKI fulfills all the criteria for a valuable tool to study TREK1, TREK2, and TRESK channel functions: (i) it is inactive in the dark or ambient light, (ii) a brief pulse of light at 365 nm induces an efficient photoblock, (iii) the block is stable for long periods in the dark or ambient light, and (iv) a brief pulse of light at 480 nm rapidly removes the photoblock.

### LAKI-induced acute TREK-TRESK closing regulates nociceptor excitability
We next addressed the functional effect on neuronal excitability of the light-dependent block and showed that alternating illumination between 365 and 480 nm increased and decreased the firing rate properties of nociceptors (Fig. 2e, f). This result validates that not only chronic[15] but also acute TREK closing is sufficient to generate nociceptor hyperexcitability. Furthermore, as primary nociceptors are the first neurons involved in the complex processing system that evokes normal and pathological pain, LAKI represents an attractive tool to remote-control pain signaling with high spatiotemporal precision but without delay or the need for genetic, viral, or surgical manipulations.

### Acute closing of TREK1, TREK2, and TRESK in freely moving animals induces pain behavior
We examined the capacity of LAKI to induce nocifensive behaviors when topically applied to the eye. The cornea is a transparent tissue highly innervated by nociceptive sensory terminals[19], making this system suitable for photopharmacological experiments on pain[6,20]. Topical application of vehicle solution or LAKI to the corneal surface of

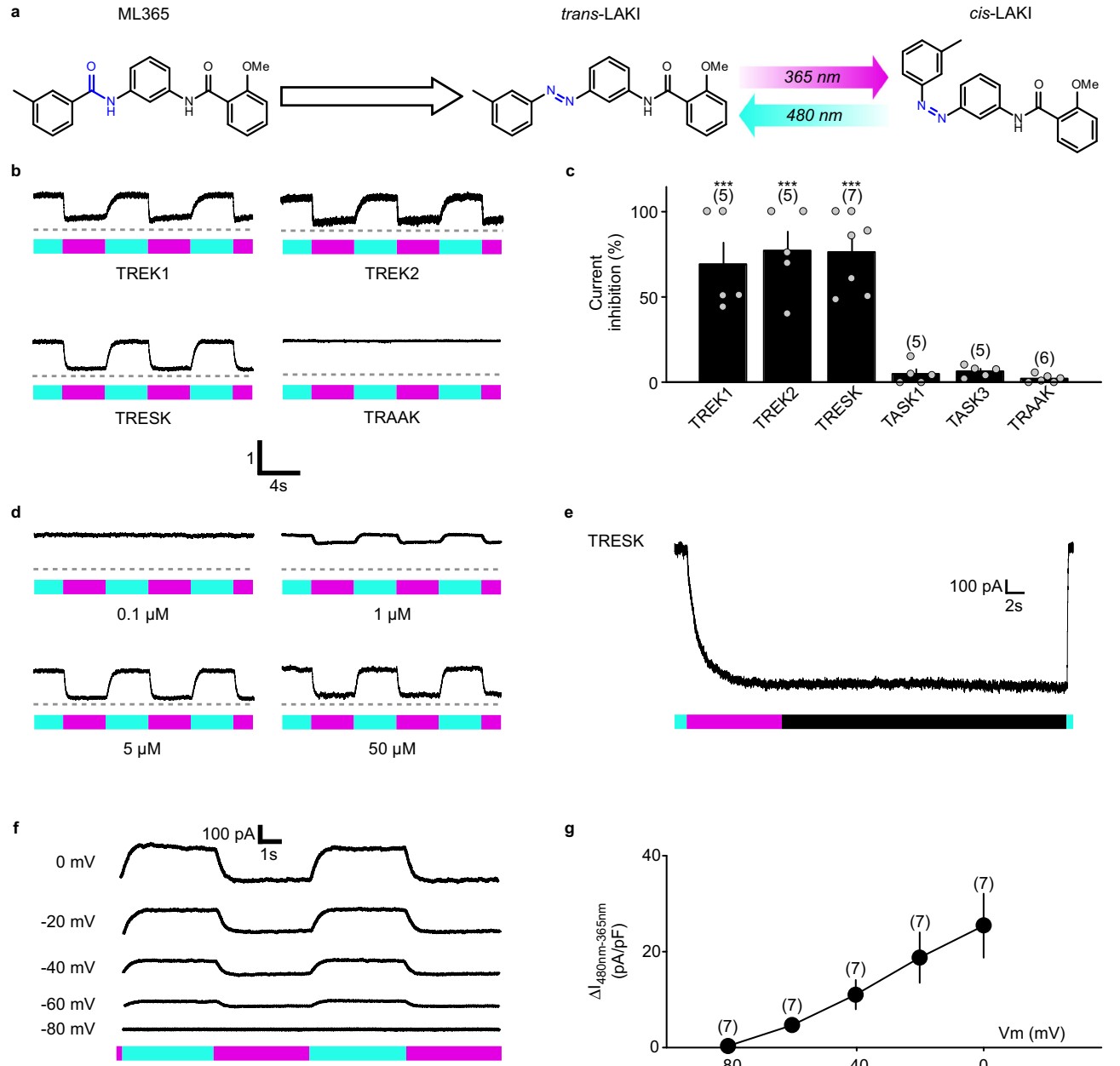

**Fig. 1 | LAKI selectively photo-controls TREK1, TREK2 and TRESK channels in a heterologous system. a** Design of LAKI from ML365 and photoisomerization of LAKI upon alternating illumination at 480 nm (blue) and 365 nm (magenta). **b** Normalized whole-cell current recording elicited at −60 mV from HEK293T cells expressing TREK1, TREK2, TRESK and TRAAK in the presence of LAKI (5 μM) upon alternating illumination at 480 nm (blue) and 365 nm (magenta). **c** Bar graph summarizing the current inhibition (%) of TREK1, TREK2, TRESK, TASK1, TASK3 and TRAAK at −60 mV. For each channel, n was obtained from one experiment. Statistical significance was determined by QuasiBinomial GLM followed by Dunnett's post-test versus TASK1 (***$p < 0.001$). **d** Normalized whole-cell current recordings elicited at −60 mV from HEK293T cells expressing TRESK in the presence of several

concentrations of LAKI upon alternating illumination at 480 nm (blue) and 365 nm (magenta). **e** Representative whole-cell current recording elicited at 0 mV from HEK293T expressing TRESK in the presence of LAKI (5 μM) upon illumination at 480 nm (blue) and 365 nm (magenta) or in the dark. **f** Whole-cell current recordings elicited at different holding potentials from HEK293T cells expressing TRESK in the presence of LAKI (5 μM) upon alternating illumination at 480 nm (blue) and 365 nm (magenta). **g** IV relationship of the photocurrent density induced by alternating illumination ($I_{480\,nm} - I_{365\,nm}$) at different holding potentials in HEK293T cells expressing TRESK in the presence of LAKI (5 μM). n was obtained from one experiment. Data were represented as mean ± SEM. The numbers of tested cells are indicated in parentheses on the graph.

the mouse eye (Fig. 3a, b) followed by a 20 s near-UV light pulse to activate LAKI, produced a ~3-fold increase of the number of nocifensive behaviors (3.83 ± 1.28 vs 11.00 ± 1.31) (Fig. 3c), specifically both scratching and wiping (considered to be itch and pain-related behaviors, respectively[21]) (Fig. 3d). Furthermore, we found that LAKI activation potentiates a painful behavior provoked by capsaicin (16.38 ± 2.45 vs 28.10 ± 2.34), a chemical stimulus that activates nociceptors (Supplementary Fig. 11), validating *cis*-LAKI activation of these

neurons. Therefore, LAKI controls both ocular acute pain and itch by inducing nociceptor activation.

### Using LAKI in well-established, classical behavioral assays: Hargreaves and Von Frey tests

As TREK-TRESK acute closing in the sensory terminals triggers spontaneous pain, we wondered if we could use LAKI to specifically target TREK-TRESK in the most commonly used stimulus-evoked pain tests,

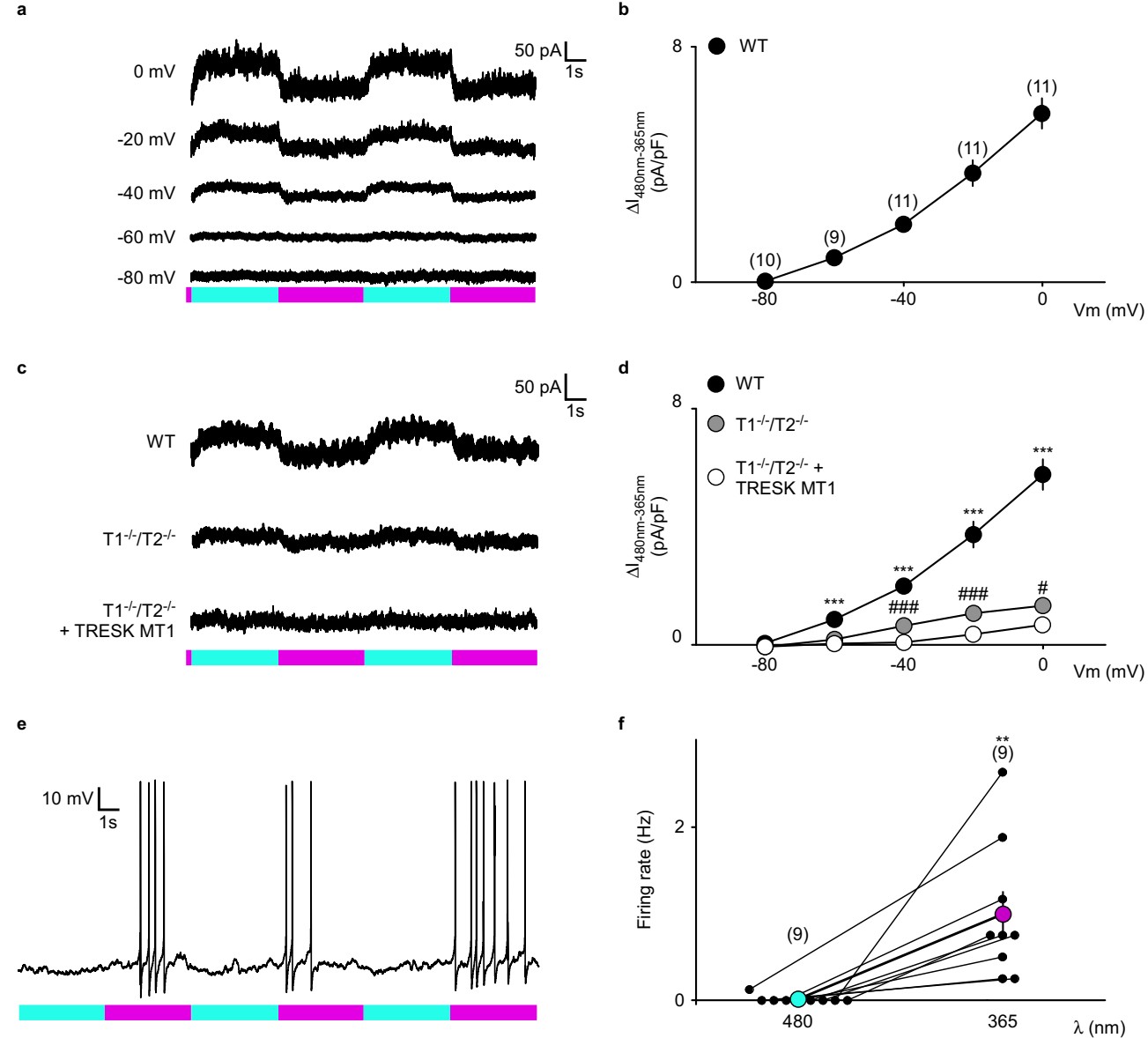

**Fig. 2 | LAKI specifically photoblocks TREK1, TREK2 and TRESK in native small sensory neurons. a** Whole-cell current recordings elicited at different holding potentials from wild-type (WT) TG neurons in the presence of LAKI (10 μM) upon alternating illumination at 480 nm (blue) and 365 nm (magenta). **b** IV relationship of the photocurrent density induced by alternating illumination ($I_{480\,nm} - I_{365\,nm}$) in WT TG neurons in the presence of LAKI (10 μM). n was obtained from six mice from two independent experiments. **c** Whole-cell current recordings elicited at 0 mV from WT TG neurons, *Trek1^(-/-)/Trek2^(-/-)* TG neurons and *Trek1^(-/-)/Trek2^(-/-)* TG neurons overexpressing TRESK-MT1 in the presence of LAKI (10 μM) upon alternating illumination at 480 nm (blue) and 365 nm (magenta). **d** IV relationship of the photocurrent density ($I_{480\,nm} - I_{365\,nm}$) in WT TG neurons, *Trek1^(-/-)/Trek2^(-/-)* TG neurons and *Trek1^(-/-)/Trek2^(-/-)* TG neurons overexpressing TRESK-MT1 in presence of LAKI (10 μM). Statistical significance was determined by a mixed-effects model with repeated measures followed by Holm–Sidak's post-test (\*\*\**p* < 0.001 compared WT

TG with *Trek1^(-/-)/Trek2^(-/-)* TG and *Trek1^(-/-)/Trek2^(-/-)* TG + TRESK-MT1; ###*p* < 0.001, #*p* < 0.05 compared *Trek1^(-/-)/Trek2^(-/-)* TG with *Trek1^(-/-)/Trek2^(-/-)* TG⁺ TRESK-MT1). The number of tested neurons is for WT *n* = 10 at −80 mV, *n* = 9 at −60 mV, *n* = 11 at −40, −20, and 0 mV from six mice from two independent experiments; for *Trek1^(-/-)/Trek2^(-/-)* *n* = 7 at −80 and −60 mV, *n* = 9 at −40, −20, and 0 mV from six mice from two independent experiments; for *Trek1^(-/-)/Trek2^(-/-)* + TRESK-MT1 *n* = 8 at −80, −60, −40, −20, and 0 mV from six mice from two independent experiments. **e** Representative voltage trace showing the firing modulation of WT TG neurons in the presence of LAKI (10 μM) upon alternating illumination at 480 nm (blue) and 365 nm (magenta). **f** Graph summarizing the average firing rate of WT TG neurons. *n* was obtained from 6 mice from two independent experiments. Statistical significance was determined by a two-sided paired *t*-test (\*\**p* = 0.0053). Data were represented as mean ± SEM. The numbers of tested neurons are indicated in parentheses on the graph.

which mimic studies of enhanced pain in humans. Notably, we looked at the ability of LAKI to trigger an enhanced response to noxious stimuli (hyperalgesia) or a nociceptive response to innocuous stimuli (allodynia).

The Hargreaves test consists of measuring the hind paw withdrawal latency following a thermal painful stimulus[22]. After LAKI injection, mice were allowed to freely explore a chamber with a transparent floor. In the absence of light, LAKI did not modify the paw withdrawal latency

compared to mice having received saline solution (*P* > 0.39), showing that LAKI is not active at rest and does not disrupt the normal physiology of mouse thermal perception (Fig. 3e). By contrast, we found that LAKI activation upon a 20 s 365 nm light pulse (<1 mW/mm²) before the thermal stimulus induced a decrease of the paw withdrawal latency (Fig. 3e). This UV light-induced thermal hypersensitivity was prevented by co-application of LAKI with ML67.33, a specific TREK channel agonist[23,24], supporting TREK channel involvement (*P* > 0.21) (Fig. 3g).

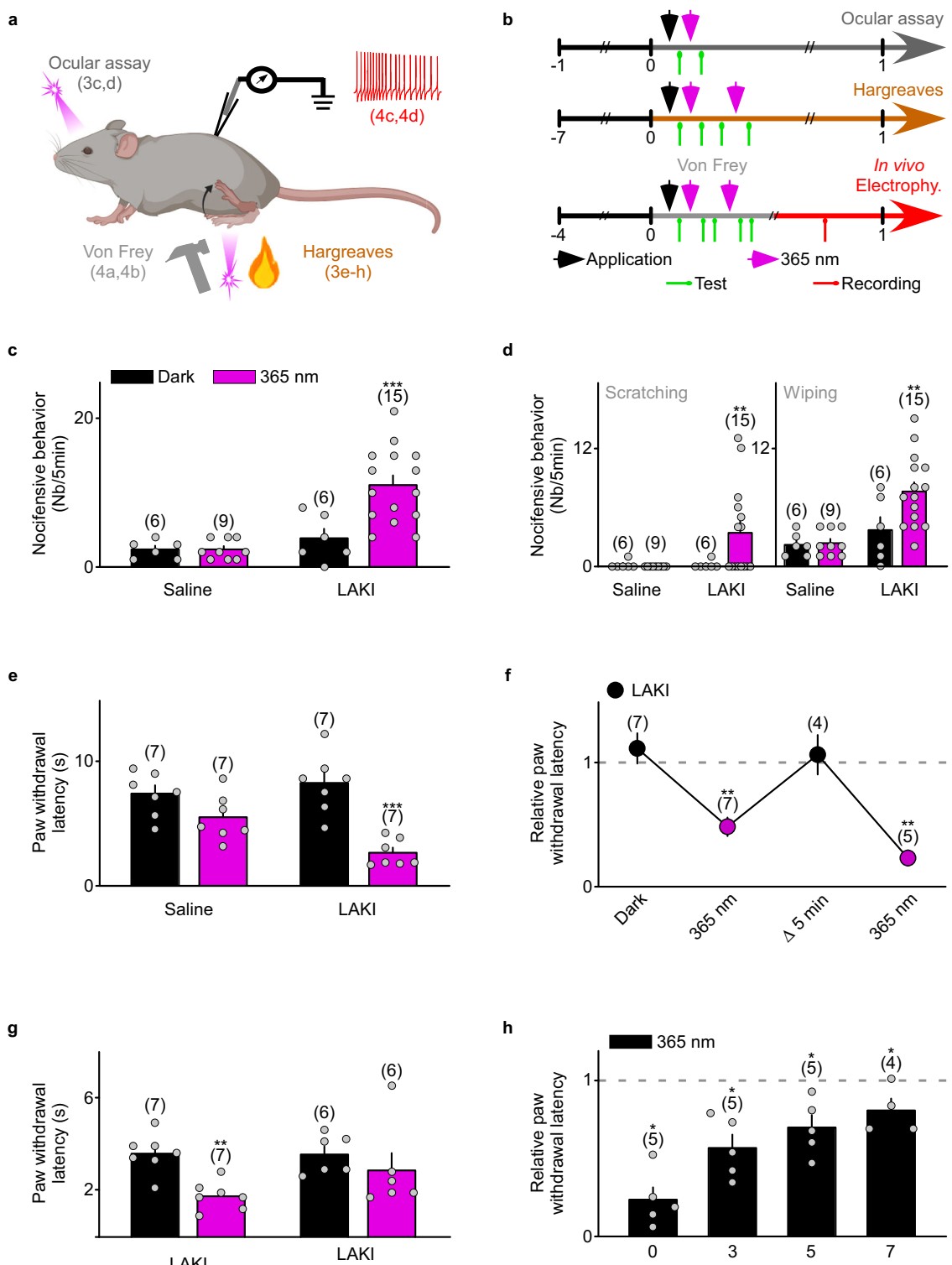

A good model for physiological studies must show quick reversibility and reproducibility of the effect. As shown in Fig. 3f, we found that the thermal hypersensitivity was fully reversed after ~5 min following the light pulse ($1.06 \pm 0.16$ vs $1.00 \pm 0.11$, $P > 0.74$) due to the relaxation of LAKI to the *trans*-state in the dark. LAKI can be reactivated for further cycles following a new pulse of light at 365 nm without any loss of effect (Fig. 3f). Finally, we assessed how long LAKI remained functional in the tissue. The UV-induced hyperalgesia could be observed more than seven days post-LAKI injection (Fig. 3h), which points to a high stability of LAKI in vivo. This is supported by our

in vitro results, where no loss of efficiency has been observed for LAKI after eight days at room temperature ($P > 0.98$) (Supplementary Fig. 12).

Next, we adapted this procedure to test mechanical pain perception using the Von Frey assay, the most routinely used test in rodents and humans to evaluate clinical mechanical allodynia[25,26]. This assay consists of measuring the required pressure to be applied to induce a paw withdrawal of the rodent (Fig. 3a, b). As expected after injection, LAKI did not modify the mechanical sensitivity at rest ($P > 0.99$) but following a

**Fig. 3 | LAKI controls pain behavior in freely moving mice. a** Representation of the experimental procedures (created with BioRender.com). **b** Schematic of experimental behavioral assays. Black arrows represent the injection or topical application of saline solution or LAKI. Magenta arrows represent the pulse of light at 365 nm. Green arrows represent the measurement of nocifensive behavior or thermal and mechanical sensitivity. The red arrow represents extracellular in vivo recording. **c** Bar graph summarizing the average of nocifensive behavior elicited by mice after ocular application of either saline or LAKI (100 µM, 5 µL) solution preceded or not by a 20 s illumination pulse (<1 mW/mm²) at 365 nm (magenta). *n* was obtained from three independent experiments. Statistical significance was determined by Poisson GLM followed by Bonferroni's post-test (***p < 0.001). **d** Bar graph showing the average of nocifensive behavior, making the distinction between scratching and wiping behavior, elicited by mice after topical application of either saline or LAKI (100 µM, 5 µL) solution preceded or not by a 20 s illumination at 365 nm (magenta). n was obtained from three independent experiments. Statistical significance was determined by Poisson GLM followed by Bonferroni's post-test (**p = 0.00564 for scratching, **p = 0.0035 for wiping). **e** Bar graph summarizing the average of the thermal paw withdrawal latency of mice injected either with saline solution or with LAKI (100 µM, 15 µL) before and after 20 s illumination at

365 nm (magenta). *n* was obtained from one experiment. Statistical significance was determined by two-way ANOVA with repeated measures followed by Holm−Sidak's post-test (***p = 0.0002). **f** Graph summarizing the average of the relative paw withdrawal latency of mice injected with LAKI (100 µM, 15 µL) relative to mice injected with saline solution. *n* was obtained from one experiment. Statistical significance was determined by a mixed-effects model with repeated measures followed by Holm−Sidak's post-test (**p = 0.0074 and p = 0.0017 respectively for first and second illumination at 365 nm). **g** Bar graph summarizing the average of the thermal paw withdrawal latency of mice injected either with LAKI (100 µM, 15 µL) or with LAKI plus ML67.33 (100 µM) after 20 s illumination at 365 nm. *n* was obtained from one experiment. Statistical significance was determined by a mixed-effects model with repeated measures followed by Holm−Sidak's post-test (**p = 0.0062). **h** Bar graph summarizing the average of the relative paw withdrawal latency at 0, 3, 5, and 7 days post-injection after 20 s illumination at 365 nm (magenta). *n* was obtained from one experiment. Statistical significance was determined by a mixed-effects model with repeated measures followed by Holm−Sidak's post-test (*p = 0.0233, p = 0.0104, p = 0.0469, and p = 0.0469 respectively for Days 0, 3, 5, and 7). Data were represented as mean ± SEM. The numbers of mice are indicated in parentheses on the graph.

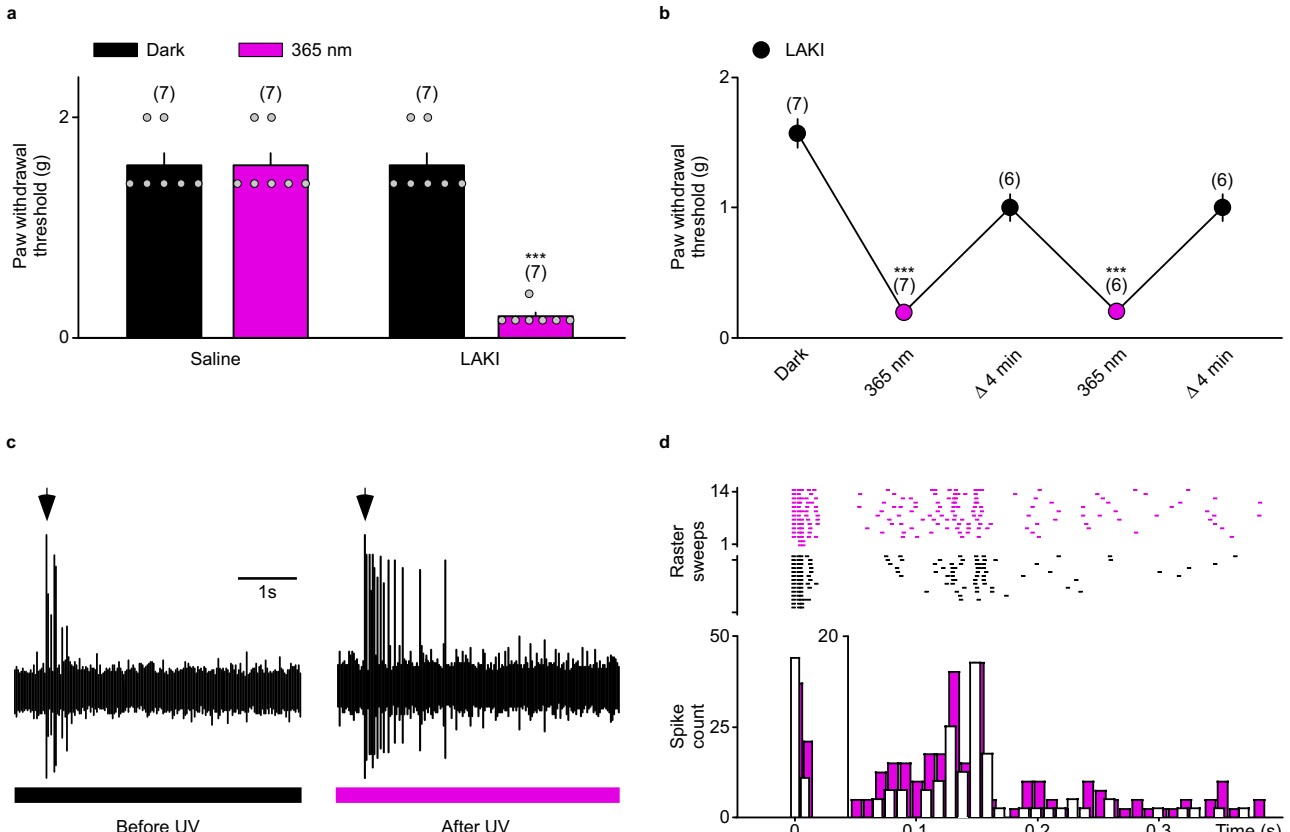

**Fig. 4 | LAKI controls mechanical allodynia in mice by sensitization of nociceptor excitability. a** Bar graph summarizing the paw withdrawal threshold of mice injected either with saline solution or with LAKI (100 µM, 15 µL) in the dark or after 20 s illumination at 365 nm (magenta). *n* was obtained from seven independent experiments. Statistical significance was determined by a mixed-effects model with repeated measures followed by Holm−Sidak's post-test (***p < 0.001). **b** Graph summarizing the average of the paw withdrawal threshold of mice injected with LAKI (100 µM, 15 µL) relative to mice injected with saline solution, in the dark or after 20 s illumination at 365 nm (magenta). *n* was obtained from seven

independent experiments. Statistical significance was determined by a mixed-effects model with repeated measures followed by Holm−Sidak's post-test (***p < 0.001). **c** Representative spinal dorsal horn neuron responses evoked by electrical stimulation (4 mA) before and after 20 s illumination at 365 nm (magenta) in mice injected with LAKI (100 µM, 15 µL). **d** Raster plot and superimposed post-stimulus histogram of dorsal horn neurons before (black) and after illumination at 365 nm (magenta) in mice injected with LAKI (100 µM, 15 µL). Data were represented as mean ± SEM. The numbers of mice are indicated in parentheses on the graph.

transdermal 20 s UV light pulse, we observed a ~6-fold decrease of the paw withdrawal threshold (Fig. 4a). Again, this hypersensitivity was reversed after ~4 min following the UV light pulse (*P* > 0.098) (Fig. 4b). Similarly, LAKI could be used for several

cycles without showing any reduction of its potency over time (Fig. 4b). This UV light-induced allodynia was prevented by *Trek1/Trek2* genetic invalidation (*P* > 0.99), further supporting TREK channel involvement (Supplementary Fig. 13). We then addressed

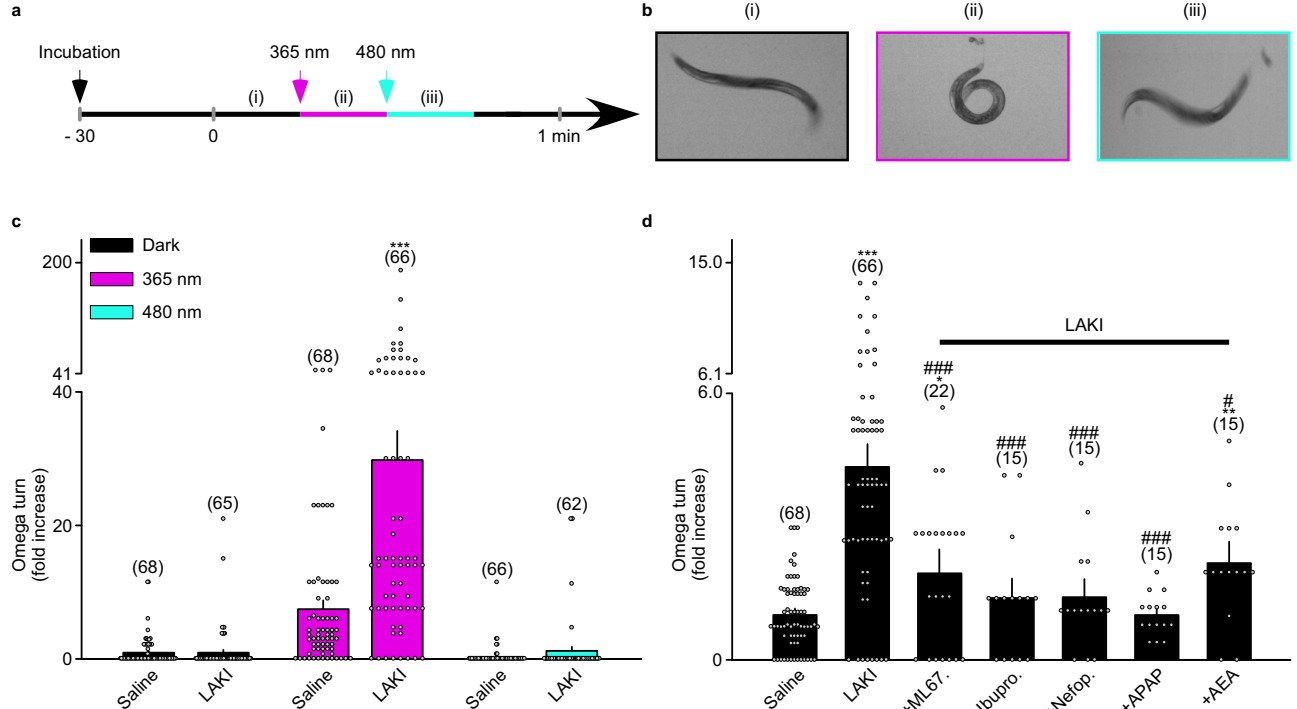

**Fig. 5 | LAKI enables optical control of conserved pain pathways in *C. elegans*.** **a** Schematic of the experimental behavioral assay. The black arrow represents the incubation of C. *elegans* in saline or 100 μM LAKI solution. The Magenta arrow represents the beginning of 15 s application of light at 365 nm. The blue arrow represents the beginning of the 15 s application of light at 480 nm. **b** Representative observed behavior in the dark (i), then during 365 nm light application (ii), and during 480 nm light illumination (iii). **c** Graph summarizing the fold increase generation of Omega turns under 365 or 480 nm illumination compared to the dark, in the presence of LAKI (100 μM) or with saline solution. n was obtained from four independent experiments. Statistical significance was determined by QuasiPoisson GLM followed by Bonferroni's post-test (***$p < 0.001$). **d** Bar graph summarizing the relative average of Omega turns made either in the presence of LAKI (100 μM), LAKI

(100 μM) plus ML67.33 (80 μM), LAKI (100 μM) plus Ibuprofen (100 μM), LAKI (100 μM) plus Nefopam (100 μM), LAKI (100 μM) plus APAP (100 μM), LAKI (100 μM) plus AEA (100 μM) or equivalent DMSO upon 15 s illumination at 365 nm. *n* for Saline and LAKI was obtained from four independent experiments, *n* for analgesic molecules was obtained from one experiment. Statistical significance was determined by QuasiPoisson GLM followed by Dunnett's post-test (*$p = 0.01499$, **$p = 0.00724$, ***$p < 0.001$ versus control; #$p = 0.01449$, ###$p = 0.000482$, $p = 0.000321$, $p = 0.000352$, and $p = 0.0000523$ respectively for ML67.33, Ibuprofen, Nefopam and APAP versus LAKI). Data are represented as mean ± SEM adjusted for the propagation of uncertainties. The numbers of *C. elegans* are indicated in parentheses on the graph.

whether this modulation was linked to the sensitization of nociceptors, similar to what we observed in vitro.

## LAKI regulates C-fiber excitability in vivo

Sensory neurons are pseudo-unipolar, with a peripheral branch that terminates in the skin and a central branch that terminates in the dorsal horn of the spinal cord. Nociceptive signals are sent to the spinal cord and brain to be integrated and felt as pain sensation[27]. To determine their activity at the single fiber level in vivo, we used the same injected mice used for the Von Frey test to investigate the effect of LAKI activation on the response of the spinal dorsal horn neurons following electrical stimulation of their receptive field. The evoked response, i.e. the number of delayed C-spikes induced by the stimulation, was increased by LAKI when activated by transdermal illumination before stimulation (Fig. 4c, d), demonstrating that the LAKI-induced hyperalgesia/allodynia is related to an increase of nociceptor excitability due to TREK/TRESK acute closing.

These results have two implications. First, as the majority of the UV-A light cannot penetrate deeper than the epidermis[28] which exclusively contains free nerve endings[29], the sensitization induced by near-UV light is therefore due to a local channel blocking. This indicates that TREK-TRESK channels are localized and functional at rest in nociceptor nerve endings and that their acute closing is sufficient to activate peripheral fibers and induce pain sensitization. Second, these results demonstrate that LAKI, by controlling endogenous channel activity regulating nociceptor excitability in physiological conditions,

is suitable to study pain processes with many advantages. Notably, immediately after injection, LAKI allows a highly reproducible photocontrol of pain, non-invasively in freely moving animals with a high spatiotemporal resolution and without the need of constant illumination.

## LAKI-induced optical control of *C. elegans* behavior

Despite the relative simplicity of their nervous system, invertebrates share with vertebrates common genetic mechanisms regulating acute and chronic nociception[30]. We therefore wondered if LAKI-induced pain is conserved among different animal classes. *C. elegans* is a suitable animal model for photopharmacological studies of $K_{2P}$ channels and nociception since it is transparent and its genome contains more than 40 $K_{2P}$ channel orthologs (notably TREK1/2 and TRESK orthologs, respectively SUP-9, twk-28, twk-39, and twk-48)[31].

We focused on a stereotypical movement named "Omega turn", which is part of the escape behavior response to stressful stimuli such as UV light[32] (Fig. 5a, b). After 30 min incubation, we found that LAKI does not alter the normal behavior of *C. elegans*. Second, we found that LAKI activation by a 365 nm light illumination (<1 mW mm$^{-2}$) induced an approximately four fold increase in the Omega turn numbers compared to control *C. elegans* (29.79 ± 4.35 vs 7.50 ± 1.32) (Fig. 5c). Third, we applied 480 nm light to turn off LAKI, inducing a total reversion of the behavior (1.27 ± 0.60 vs 0.35 ± 0.19, $P > 0.12$) (Fig. 5c). This behavior could be partially prevented by co-applying ML67.33 with LAKI (4.32 ± 0.38 vs 1.94 ± 0.36) (Fig. 5d), supporting the

involvement of the TREK orthologs in this hypersensitivity. Furthermore, the co-application of Ibuprofen, an analgesic drug, drastically reduced the number of Omega turns (4.32 ± 0.38 vs 1.38 ± 0.36), supporting the relevance of the model for pain functional studies (Fig. 5d). This simple model may be used to develop an in vivo High Throughput Screening (HTS) method in which the read-out is the *C. elegans* behavior, to find TREK agonists or more general analgesic drugs. To validate this hypothesis, we tested several compounds. Whereas co-application of Capsaicin with LAKI did not modify the escape behavior of the worm (Supplementary Fig. 14), Nefopam, Paracetamol, and Anandamide drastically reduced the number of Omega turns compared to *C. elegans* incubated with LAKI (Fig. 5d). This analgesic-reversed effect is related to LAKI since it is not observed in its absence (Supplementary Fig. 15).

Taken together, these results demonstrate that TREK/TRESK channel orthologs may be involved in the escape behavior of *C. elegans*. This behavior can be considered nociception-related because UV light is deadly for worms and the response is reduced by analgesics. This demonstrates the conservation of the function of $K_{2P}$ channels in pain signaling within the animal kingdom as observed for other ion channels such as TRP channels[33]. More importantly, we found that the LAKI sensitization of worms to UV light-induced Omega turns can be used as an easy read-out for the development of a fast, inexpensive, and robust in vivo HTS method for analgesic drug discovery.

## Discussion

To summarize, we synthesized LAKI, an easy, stable, and specific light-activated inhibitor of TREK1, TREK2 and TRESK channels. This photopharmacological tool enabled us to functionally map TREK/TRESK channels at the free nerve endings of both TG and DRG nociceptors. Importantly, we found that local acute channel closing in nerve endings generates instantaneous pain as well as hyperalgesia or allodynia, demonstrating the key role of TREK/TRESK in pain. Furthermore, LAKI enabled us to demonstrate that TREK/TRESK orthologs present in other species such as *C. elegans* are transducers in conserved pain perception pathways. More importantly, thanks to the localization and function of TREK-TRESK and LAKI light sensitivity, LAKI can be used to remotely control pain with many advantages. With only a simple injection or incubation, LAKI endows a reversible and reproducible light-control of nociception in different freely moving animal models, for several days, without the need for transfection, infection, genetic manipulation, or surgery procedures, and is adaptable to several species. Moreover, being inert in the dark and stable LAKI is not invasive, making it, with its stability in vivo for at least one week and bi-stability upon UV light illumination, compatible with long-term experiments[34,35]. These features make LAKI an appropriate tool in accordance with animal welfare rules by reducing animal housing and distress caused by existing methods for pain induction, and by improving the repeatability and reducing variability (the same animal being the control and the test).

Finally, LAKI is ideal for basic and translational pain research, providing straightforward and reproducible control of pain with spatiotemporal resolution in freely moving animals, and for in vivo analgesic high throughput drug screening in worms before validation in mammals.

## Methods

### General experimental details for chemical synthesis

**Synthesis.** Reactions were performed using chemicals obtained from Alfa Aesar, Sigma-Aldrich, VWR, and Thermo Scientific. All dry reactions were performed in Extra Dry Solvents obtained from Acros Organics under a nitrogen atmosphere and in oven-dried glassware. Reactions were monitored using thin-layer chromatography on silica plates 60Å (250 $\mu$m) from Silicycle and were observed under a UV lamp and stained with Iodine. All reactions were washed and/or quenched, followed by multiple extractions with ethyl acetate or methylene chloride in a separatory funnel. The combined extracted organic layers were dried over anhydrous sodium sulfate, filtered, and then concentrated under a vacuum on a rotary evaporator. Unless otherwise noted, all reactions were purified by flash chromatography following Still's procedure with 60Å (40–63 um) Silica Gel obtained from Silicycle. All NMR spectra were measured in deuterated chloroform ($CDCl_3$) and acetonitrile ($CD_3CN$) with Varian (Innova) or Bruker (Neo) spectrometers, at 400 or 500 MHz for $^1H$ spectra and 101 or 151 MHz for $^{13}C$ spectra. Spectra were calibrated to residual solvent peaks as reported by Fulmer et al.[36]. UV-Visible absorption spectroscopy was performed using a Vernier UV-VIS Spectrophotometer and the spectra were analyzed using Logger Lite 1.9.4. HR-MS was obtained in ESI mode from the Mass Spectrometry Lab operated by the School of Chemical Science at the University of Illinois Urbana-Champaign.

**Synthesis of 2.** About 500 mg (6.62 mmol) of 3-nitroaniline **1** was dissolved in 22 ml of dichloromethane (DCM), and to that, a solution of 4.45 g (7.24 mmol) of Oxone® (monopersulfate compound) in 20 ml of water was slowly added under continuous stirring. The reaction mixture was stirred at room temperature for 12 h until completion, at which point it was diluted with 50 ml of DCM and washed with sodium bicarbonate (50 ml × 3), brine (50 ml × 2), dried over sodium sulfate, and concentrated under vacuum to yield a green nitroso product (**2**). The concentrated product was used directly without further purification.

**Synthesis of 4.** About 602 mg (3.96 mmol) of nitroso **2** was dissolved in 30 ml of ethyl acetate with 10 ml of glacial acetic acid and stirred at room temperature for 10 min under an $N_2$ atmosphere. To this 0.44 ml (3.96 mmol) of m-toluidine **3** was added dropwise over 5 min. The reaction was stirred at room temperature for 22 h till reaction completion. The reaction was diluted with 30 ml of ethyl acetate and quenched with 25 ml of 1 M NaOH, followed by washing with sodium bicarbonate (50 ml × 3), brine (50 ml × 2), and dried over sodium sulfate. The product was purified via column chromatography using hexanes: ethyl acetate = 5:0.1 to yield 601.3 mg (71.9%) of a red solid **4**.

$R_f$ = 0.47 (hexanes: ethyl acetate = 5:0.1)

**HR-MS** (ESI): m/z calculated [MH]$^+$ for $C_{13}H_{12}N_3O_2$ is 242.0930, found [MH]$^+$ as 242.0925.

$^1H$ **NMR** (DMSO, 400 MHz): δ = 8.46 (t, J = 1.9 Hz, 1H), 8.34 (ddd, J = 8.2, 2.3, 1.0 Hz, 1H), 8.27 (qd, J = 7.9, 3.2, 1.0 Hz, 1H), 7.83 (t, J = 8.2 Hz, 1H), 7.71 (s, 2H), 7.47 (d, J = 7.4 Hz, 1H), 7.41 (d, J = 7.6 Hz, 1H), 2.39 (s, 3H).

$^{13}$C NMR (DMSO, 101 MHz): δ = 152.1, 151.6, 148.6, 139.1, 133.1, 131.1, 129.8, 129.3, 125.3, 122.9, 120.7, 115.4, 20.8.

**Synthesis of 5.** About 539 mg (2.55 mmol) of **4** and 1.84 g (7.66 mmol) of $Na_2S$ were added to the reaction vessel and flushed with $N_2$. To this, 8 ml of Dioxane was added under constant stirring and then heated to 90 °C for 4 h followed by 12 h at rt. The reaction did not go to completion, so another 1.84 g of $Na_2S$ was added and the reaction was reheated for 4 h at 90 °C, followed by stirring at room temperature for another 12 h. The reaction was diluted with 30 ml of ethyl acetate followed by washing with Brine (50 ml × 4) and drying over sodium sulfate. The product was purified via column chromatography using hexanes: ethyl acetate: DCM = 3:1:1 to yield 449.8 mg (83%) of red oil **5**.

$R_f$ = 0.69 (hexanes: ethyl acetate: DCM = 3:1:1)

**HR-MS** (ESI): m/z calculated [MH]$^+$ for $C_{13}H_{14}N_3$ is 212.1188, found [MH]$^+$ as 212.1185.

$^1$H NMR (CDCl$_3$, 400 MHz): δ = 8.72 (t, J = 1.9 Hz, 1H), 8.30 (dd, J = 7.6, 3.1 Hz, 1H), 8.23 (dd, J = 7.9, 2.5 Hz, 1H), 7.78 (s, 2H), 7.70 (t, J = 8.4 Hz, 1H), 7.44 (t, J = 8.1 Hz, 1H), 7.36 (d, J = 7.7 Hz, 1H), 2.48 (s, 3H).

$^{13}$C NMR (CDCl$_3$, 101 MHz): δ = 153.1, 152.3, 149.1, 139.3, 133.1, 130.0, 129.3, 129.2, 124.9, 123.5, 121.0, 117.1, 21.4.

**Synthesis of LAKI.** About 283 mg (1.34 mmol) of **5** was dissolved in 12 ml of THF under an $N_2$ atmosphere and cooled to 0 °C. To this 0.7 ml (4.02 mmol) of diisopropylethylamine was added and stirred at 0 °C for 5 min. 0.24 ml (1.60 mmol) of o-anisoyl chloride **6** was added dropwise over 10 min. The reaction was stirred at 0 °C for 4 h until reaction completion, at which point the reaction was diluted with 50 ml of ethyl acetate followed by washing with sodium bicarbonate (40 ml × 5), brine (50 ml × 2), and dried over sodium sulfate. The product was purified via column chromatography using hexanes: ethyl acetate = 6:1 to yield 405.2 mg (86.3%) of red solid LAKI.

$R_f$ = 0.37 (hexanes: ethyl acetate = 3:0.5)

**HR-MS** (ESI): m/z calculated [MH]$^+$ for $C_{21}H_{20}N_3O_2$ is 346.1556, found [MH]$^+$ as 346.1554.

$^1$H NMR (CD$_3$CN, 500 MHz): δ = 9.90 (s, 1H), 8.35 (s, 1H), 8.10 (d, J = 7.4, 1.8 Hz, 1H), 7.78 (d, J = 8.1 Hz, 1H), 7.75 (s, 1H), 7.71 (d, J = 8.1 Hz, 1H), 7.66 (d, J = 8.3 Hz, 1H), 7.56 (q, J = 8.3, 7.6 Hz, 2H), 7.46 (t, J = 7.6 Hz, 1H), 7.38 (d, J = 7.4 Hz, 1H), 7.19 (d, J = 8.4 Hz, 1H), 7.13 (t, J = 7.5 Hz, 1H), 4.05 (s, 3H), 2.46 (s, 3H).

$^{13}$C NMR (CD$_3$CN, 151 MHz): δ = 164.7, 158.5, 154.0, 153.6, 140.7, 140.4, 134.4, 133.0, 132.5, 130.6, 130.1, 124.0, 123.8, 123.0, 122.1, 120.9, 119.7, 114.7, 113.0, 57.0, 21.3.

## Photoswitching procedure in NMR

A solution of LAKI (440 μM) was prepared in a quartz NMR tube with Chloroform (CDCl$_3$) and sealed. The prepared sample was kept in the dark to relax for 5 days. A $^1$H-NMR was taken in the dark in a Bruker (Neo) 500 MHz NMR to record the thermally relaxed photostationary state. Alternating NMRs were taken after irradiation with LED lamps of UV$_{365nm}$ (8.7 mW/m$^2$) and Blue$_{460nm}$ (12.6 mW/m$^2$) light from a distance of 1 cm for 15 min each.

## Photoswitching procedure in UV-Vis

A solution of LAKI (5 μM) was prepared in a quartz cuvette by diluting a 5 mM solution in DMSO with HEPES (10 mM) and sealed. The sample was irradiated alternatively with LED lamps of UV$_{365 nm}$ (8.7 mW/m$^2$) and Blue$_{460nm}$ (12.6 mW/m$^2$) light for 5 min each from a distance of 1 cm. UV-Visible absorbance was measured with a Vernier UV-Vis spectrophotometer.

## Procedure for calculating thermal half-life $t_{1/2}$

To measure the thermal relaxation (*cis* to *trans* isomerization) of a solution of LAKI (5 μM in HEPES solution containing 0.1% DMSO), the solution was irradiated with UV$_{365nm}$ light to achieve the PSS$_{365}$. LAKI was allowed to thermally relax at 21.5 °C and was monitored at 330 nm for 20 h. Absorbance was plotted vs time and thermal half-life was calculated using $t_{1/2} = 0.693/k$. The Solver add-on in Microsoft Excel was used to find a line fit using the GRG nonlinear engine using the equation $Abs(t) = Abs_{BI} + (Abs_{365nm} - Abs_{BI})*\exp(-kt)$, where k is the rate constant (h$^{-1}$), $t$ is time (h), Abs$_{BI}$ is absorbance before illumination, and Abs$_{365 nm}$ is the absorbance at PSS$_{365}$. The variables Abs$_{BI}$, Abs$_{365nm}$, and k were used as free parameters in the fit.

## HEK cell culture

HEK (human embryonic kidney) 293 T cells were purchased from ATCC (CRL-11268) and maintained at 37 °C in 5% CO$_2$ in high glucose DMEM containing 10% fetal bovine serum and used from passage 10 to 40. One passage per week was made on 35 mm diameter dishes.

DNA encoding K$_{2P}$S and K$_v$ channels was cloned in the pIRES2EGFP vector. HEK293T cells were maintained in DMEM with 10% FBS on plastic dishes. Cells were transiently transfected using calcium phosphate with 3.6 μg of DNA. When two genes were co-expressed, a ratio of 1:1 DNA was used. TG neurons were transfected with 0.75 μg of DNA using JetPRIME.

## Primary culture of mouse TG neurons

Trigeminal ganglion tissues were collected from postnatal day 1–10 mice of either sex and treated with a mix of 1 mg/ml collagenase type II (Gibco) and BSA for ~45 min, followed by 5 mg/ml trypsin for 10 min. Neurons were dissociated by triturating with fire-polished glass pipettes and seeded on poly-lysine and Laminin coated coverslips. The DMEM/F12-based culture medium contained 2 mM L-glutamine, 10% fetal bovine serum, 100 ng/ml neural growth factor, and penicillin/streptomycin.

## Electrophysiology

HEK293T cells were recorded 24–48 h after phosphate calcium transfection (with 3.6 μg DNA). For co-expression of KCNQ1 and KCNE1 a DNA ratio of 1:1 was used. Glass pipettes were pulled with a resistance

<5 MΩ and filled with intracellular solution containing (in mM): 155 KCl, 5 EGTA, 3 MgCl₂, 10 HEPES, pH 7.3 with KOH. Cells were patch clamped using a MultiClamp 700B (Molecular Devices) amplifier, recorded using pCLAMP 11, in an extracellular solution containing (in mM): 150 NaCl, 5 KCl, 2 CaCl₂, 10 HEPES, pH 7.4 with NaOH. Currents were elicited in voltage-clamp mode with voltage-ramps (from −100 to 100 mV, 500 ms in duration). Photocurrents were elicited in voltage-clamp mode either with voltage steps (from −80 to 80 mV, 8 s in duration) or in gap-free at several holding potentials upon alternating illumination at 480 nm (blue) and 365 nm (magenta) (both illuminations 2 or 4 s in duration, respectively).

Photomodulation of neuronal excitability and photocurrent were studied in small-diameter TG neurons. *Trek1⁻/⁻;Trek2⁻/⁻* TG neurons were transfected with 0.75 μg of the pIRES2EGFP vector containing the TRESK-MT1 insert, in which there is no N-terminal tag on the insert and EGFP is co-translated as a transfection marker, or the pIRES2EGFP control plasmid with JetPRIME. The extracellular solution contained (in mM): 140 NaCl, 5 KCl, 1 MgCl₂, 2 CaCl₂, 10 HEPES, 10 glucose, pH 7.4 with NaOH. The intracellular solution used was the same used for HEK293T cells. Recording pipettes had <5 MΩ resistance. Series resistance (<20 MΩ) was not compensated. Signals were filtered at 10 kHz and digitized at 20 kHz. After establishing whole-cell access, membrane capacitance was determined with amplifier circuitry. The amplifier was then switched to current-clamp mode to measure resting membrane potential (Vrest). Neurons were excluded from analysis if the Vrest was higher than −40 mV. To assess photoswitched neuronal excitability, the depolarizing current was injected to reach the threshold for triggering action potentials then neuronal excitability was studied upon alternating illumination at 480 nm (blue) and 365 nm (magenta). The amplifier was then switched to voltage-clamp mode to assess neuronal photocurrent at several holding potentials upon the same alternating illuminations.

Concentrations of LAKI and all compounds used are indicated in the respective figure legends.

### Mouse strains

Mice lacking *Trek1* and *Trek2* were generated as described in ref. [37]. Null mutations were backcrossed against the C57BL/6 J inbred strain for 10+ generations prior to establishing the breeding cages to generate subjects for this study. Age- and sex-matched C57BL/6 J WT mice, aged 9–12 weeks, were obtained from Charles River Laboratories (Wilmington, MA).

All mouse experiments were conducted according to national and international guidelines and have been approved by the ethical committees (Ministère français de la Recherche, de l'Enseignement et de l'Innovation; University of Barcelona, CEEA, Generalitat de Catalunya, #129/21). The C57BL/6 J breeders were maintained on a 12 h light/dark cycle with constant temperature (21–23 °C), humidity (45–50%), and food and water ad libitum at the animal facility of Valrose or the Medical School of the University of Barcelona.

### Thermal sensitivity measurements

Two groups of mice were injected subcutaneously into the plantar surface of the hind paws either with 15 μl of LAKI (100 μM) or the vehicle (1% DMSO). Mice were placed individually in compartments of Plantar test Hargreaves Apparatus (Ugo Basile, 37370) for 10 min for habituation. Mice were trained for 7 days before the experiments. The thermal withdrawal latency of mice was determined by exposition to an infrared source (intensity of 50%) before and after UV exposition. The hind paws were exposed to UV by illumination with a 365 nm lamp for 20 s. Recovery of the thermal sensitivity was also evaluated 5 min after UV exposition.

### Mechanical sensitivity measurements

Mice were injected subcutaneously in the plantar surface of the hind paws with 15 μl of LAKI (100 μM) or the vehicle (1% DMSO) for the contralateral side. Mice were placed in the testing cage (Ugo Basile) for 15 min for habituation. The withdrawal threshold was determined with von Frey filaments (in g: 0.07, 0.16, 0.4, 0.6, 1, 1.4, and 2) using the ascending method (Abboud et al., 2021) before and after UV exposition. The hind paws were exposed to UV by illumination with a 365 nm laser for 20 s and the mechanical assessment was performed after a delay of 20 s. The animals were re-exposed to UV every 2 min until the end of the test. Recovery of the mechanical sensitivity was also evaluated 4 min after UV exposition. Each Von Frey filament was applied five successive times (with a delay of 30 s) on the plantar surface of the hind paw of the mouse standing on its four paws. The withdrawal threshold was considered when the mouse responded positively to three out of five applications.

### In vivo extracellular recordings

In vivo recordings were performed on mice injected with LAKI in the hind paw following the Von Frey test. Mice were anesthetized with urethane 20% (1.5 g/kg) and placed on a stereotaxic frame (Unimécanique, Asnières, France). A laminectomy was performed on lumbar vertebrae L1–L3 and segments L4–L5 of the spinal cord were exposed. Extracellular recordings of wide dynamic range dorsal horn neurons (Aby et al., 2018) were made with borosilicate glass capillaries (2 MΩ, filled with NaCl 684 mM) (Harvard Apparatus, Cambridge, MA, USA). The signal was amplified and high pass filtered using a DAM80 amplifier (WPI, FL, USA) connected to CED1401 (CED, UK). The acquisition was performed using spike 2 software (CED, UK). The criterion for the selection of a neuron was the presence of an A fiber-evoked response (0-80 ms) followed by a C fiber-evoked response (80 to 300 ms) to electrical stimulation of the corresponding receptive field of the ipsilateral paw with subcutaneously implanted bipolar electrodes connected to a stimulator (AMPI, Israel). LAKI-injected hind paw was exposed to a UV laser for 20 s every 2 min until the end of the recording. In the same experiment, a period of at least 15 min without UV was respected for recovery between two neuronal recordings.

### Ocular nocifensive behavior measurements

To assess mice nocifensive ocular responses, 5 μL of vehicle (DMSO 0.1%) or LAKI (100 μM) in saline solution, with or without Capsaicin (100 μM), were topically applied to the anterior ocular surface of the eye. Mice were lightly restrained (by grasping the scruff between the thumb and forefinger) so that the solution remained on the corneal surface. With the animal restrained, 20 s of UV light stimulation to the ocular surface was applied to activate LAKI. The number of nocifensive behaviors such as wipes (forepaw; indicative of pain) and scratches (hind paw; indicative of itch) directed to the treated eye were counted over a 5-min period. In control experiments with no UV light stimulation, the procedure was the same but without applying the light source.

### *C. elegans* strain and maintenance

Our analysis included the *C. elegans* N₂ wild-type reference strain. *C. elegans* were maintained on NGM agar plates (55 mm Petri dishes, 1.7% agar) carrying a lawn of *E. coli* OP50[38,39]. Animals were grown at 20 °C unless indicated otherwise and wild-type strain was freshly thawed prior to experiments.

### Omega turn quantification in *C. elegans*

Hermaphrodite *C. elegans* were included in the analysis at the adult stage. *C. elegans* were transferred on 35 mm diameter dishes containing M9 buffer supplemented either with LAKI or equivalent DMSO for 30-min incubation in the dark. *C. elegans* were recorded while exposed to ambient light, UV light (365 nm, SV005, Alonefire, China), and blue light (480 nm, SV004, Alonefire, China) for 15 s each in duration. The movies of *C. elegans* were acquired using Micro-Manager 1.4 and analyzed using Fiji software. Omega turns were counted manually and were included in the analysis when

*C. elegans* fully circled. Concentrations of LAKI used are indicated in the respective figure legends.

### Quantification and statistical analysis

Analysis of obtained currents in whole-cell recordings was performed using ClampFit. Data were treated and analyzed using Excel, SigmaPlot 11.0, GraphPad Prism 8, and R according to their probability distributions. QuasiBinomial GLM was used for current inhibitions while Poisson and QuasiPoisson GLMs were used for Omega turn and nocifensive behavior counting. Neuronal excitability-, photocurrent-, current density-, paw withdrawal latency, and paw withdrawal threshold-related data were assessed to verify the assumption of Normality either with the Shapiro test or with a Q-Q plot. Data verifying Normality were analyzed using either paired *t*-test when comparing two paired conditions or two-way ANOVA RM and mixed-effects model RM when comparing more than two paired conditions; otherwise, data were analyzed using Wilcoxon signed-rank test. When more than two conditions were analyzed, the tests described above were followed by Bonferroni's, Holm–Sidak's, or Tukey's post hoc test for all pairwise comparisons or by Dunnett's post hoc test for multiple comparisons to a control group.

### Reporting summary

Further information on research design is available in the Nature Portfolio Reporting Summary linked to this article.

## Data availability

The main data generated in this study are provided in the Source Data file. Source data are provided with this paper.

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

## Acknowledgements

We thank Pr. Daniel L. Minor for providing us with the selective activator of TREK channels ML67-33 and Dr. Christian Braendle for providing us with *C. elegans*. This work was supported by a grant to G.S. by the Fondation pour la Recherche Medicale (Equipe labellisée FRM 2017, FRM DEQ20170336753) the Agence Nationale pour la Recherche (AT2R-TRAAK-Bioanalgesics ANR-17-CE18-0001 and Viral_Rhodopsins, ANR-19-CE11-0026-01), the Laboratory of Excellence "Ion Channel Science and Therapeutics" (grant ANR-11-LABX-0015-01), and the French government, through the UCAJEDI Investments in the Future project managed by the National Research Agency (ANR) with the reference number ANR-15-IDEX-01, by grant to F.L. (Equipe labellisée FRM 2020, FRM EQU202003010587 to X.G. PID2020-119305RB-I00 funded by MCIN/AEI/ 10.13039/501100011033 and Instituto de Salud Carlos III of Spain, Maria de Maeztu MDM-2017-0729 and CEX2021-001159-M (to Institut de Neurosciences, Universitat de Barcelona), Generalitat de Catalunya 2021SGR00292.

## Author contributions

A.L.-W., A.A.C., M.A.K., and G.S. contributed to the design of experiments, preparation, and revision of the article. A.L.-W., A.K., A.D., A.A.C., L.E., G.C., M.B., S.H., B.W., A.B., P.F., L.M., X.G., and E.B.-G. performed experimental studies. A.L.-W., A.A.C., M.A.K., and G.S. contributed to the data analyze and writing the paper. Resources: FL. Funding acquisition: GS. Project administration: GS

## Competing interests

The authors declare no competing interests.

## Additional information

[1]Université Côte d'Azur, CNRS, INSERM, iBV, Nice, France. [2]Laboratories of Excellence, Ion Channel Science and Therapeutics, Nice, France. [3]Fédération Hospitalo-Universitaire InovPain, Cote d'Azur University, University Hospital Center, Nice, Provence-Alpes-Côte d'Azur, France. [4]University of Maine Department of Chemistry, 178 Munson Rd., Orono, ME 04473, USA. [5]Univ. Bordeaux, CNRS, IMN, UMR 5293, F-33000 Bordeaux, France. [6]Univ Angers, Nantes Université, INSERM, Immunology and New Concepts in ImmunoTherapy, INCIT, UMR 1302, Angers, France. [7]Neurophysiology Laboratory, Department of Biomedicine, Medical School, Institute of Neurosciences, Universitat de Barcelona, c. Casanova 143, 08036 Barcelona, Spain. [8]Institut d'Investigacions Biomediques August Pi i Sunyer (IDIBAPS), c. Villarroel 170, 08036 Barcelona, Spain. [9]Centre National de la Recherche Scientifique, Institut de Pharmacologie Moléculaire et Cellulaire, Labex ICST, Université Côte d'Azur, INSERM, Valbonne, France. [10]University of Connecticut Department of Chemistry, 55 N. Eagleville Rd Storrs, Mansfield, CT 06269, USA. [11]These authors jointly supervised this work: Michael A. Kienzler and Guillaume Sandoz.
✉e-mail: michael.kienzler@uconn.edu; sandoz@unice.fr

