## [Peer Review File · Nature Communications]

A photoswitchable inhibitor of TREK channels controls pain in wild-type intact freely moving animalsEditorial Note: Parts of this Peer Review File have been redacted as indicated to remove third-party material where no permission to publish could be obtained.

REVIEWER COMMENTS

Reviewer #1 (Remarks to the Author):

Note that I have read the entire draft including supplemental materials, my expertise is in the sensory biophysics of *C. elegans*. Therefore, I have no technical comments on the sections outside of this scope.

This is a well written and convincing paper on the development and testing of a photoswitchable inhibitor (LAKI) of a nociceptive-related 2-pore-domain potassium channel. In addition to tests at the chemical and neurophysiological level, the authors test LAKI at the behavioral level in standard nociceptive/pain assays using mice and *C. elegans*. Overall I see this work as thorough and convincing. I have to leave technical comments on the chemistry and mouse assays to other reviewers. Overall I believe this is an important contribution to "pain" research assays and will be of interest to a broad audience.

Comments

Re: *C. elegans* assay

1) Experimental details are missing. Currently it is not possible to reproduce the data in the *C. elegans* experiment. a) how are the worms illuminated with the 365 nm and 480 nm light. b) what are the light sources (LED or laser) and what power is used for the 480 nm light. Identifying the light sources are important because the spectrum of the LED is different depending on the manufacturer.

2) Proper controls may be missing for the pharmacological results. It is true that omega turn responses (with LAKI) are reduced when certain analgesics are applied. It is not clear if the analgesics themselves are modifying basal responses in a way to suppress omega turns. No control data is included.

3) Fig 4. The figure legend states that data are represented as "mean \pm SEM" but what about "fold" changes. Errors were propagated using standard methods?

4) L623. "...to full light." What does "full light" indicate? Full power?

5) L624. when *C. elegans* "fully annealed." What is meant by "fully annealed" here?

6) Reference [30] is incomplete.

Minor:

L615. "*E. coli*" should be italicized.

re: rate limiting step(s). I think many people would be interested in temporal limitation of the technique and perhaps it would be useful to see a magnified view (in time) of Fig 1b in order to get a better sense of the temporal constraints in de-activation and activation. A comment on the rate of photoisomerization vs functional modulation of neuronal activity would be useful.

Reviewer #2 (Remarks to the Author):

In this study, Landra-Willm and colleagues design a photoswitchable inhibitor of K2P TREK and TRESK

channels, named LAKI, and show that it reversibly controls animal behavior in response to different painful stimuli. The authors synthesized LAKI by an "azologization"-based approach, and found that the compound is inactive in vitro in the dark, but active at 365 nm light by inhibiting selected K2P channels (TREK1, TREK2 and TRESK). They next used LAKI in vivo, in both mice and *C. elegans*, to show reversible optical control of pain or escape behavior, thus demonstrating the usefulness of their tool in different species. Finally, they provide in vivo evidence by extracellular recordings that LAKI regulates C-fiber excitability in mice. The main advance of LAKI is its inactive behavior in the dark, as compared to previously reported photochromic ligands which lock ion channels in ambient light. Overall, I find the study well executed and technically sound. However, my main concern is the lack of evidence that TREK/TRESK are effectively targeted in vivo by LAKI. Although the authors provide pharmacological evidence with ML67.33, it is not sufficient to support the claim that local acute channel closing of TREK/TRESK in mice is sufficient to induce pain behavior. In addition, the following comments need to be addressed.

Comments:

1. The authors provide strong evidence that TREK1 and TREK2 are targeted by LAKI in vitro, as they report 80% decrease of light-blocked current in T1-/-/T2-/- TG neurons. However, they do not really demonstrate this is the case in vivo (e.g. in mice), although they used ML67.33 to pharmacologically prevent light-induced thermal hypersensitivity in wild-type mice. To further support the contribution of these channels, the authors should show the absence of in vivo effects of LAKI in T1-/-/T2-/- mice.
2. The chemical structure of LAKI only differs from that of ML365 by a relatively small chemical substitution (amide versus diazene). However, the pharmacological profile of LAKI seems completely different from that of ML365 in respect to TASK1 and TASK3 channels (no inhibition induced by LAKI for TASK1 and TASK3, whereas ML365 is a full inhibitor of TASK1 and TASK3). How is it possible? Please comment.
3. Photochemical data show that the half-life of cis-LAKI at 21.5°C was 0.7 hours (i.e. 42 min). However, in vivo data obtained in mice (under both thermal and Von Frey assays) rather suggest a complete relaxation process in less than 5 minutes. How can the authors explain this difference? Is it related to temperature difference (21.7°C in vitro versus 37°C in vivo)? It would be good to measure half-life of LAKI at 37°C.
4. In line 123, can the authors provide the actual inhibition level in native nociceptors? Is that inhibition level comparable to that obtained in the recombinant system (about 80%)? In Fig. 2a, please add the zero-current level.
5. In Figs. 1f, 2a and 2c, why is there an "off-inhibition" effect under the first irradiation at 480 nm? Was the cell previously irradiated at 365 nm before recording?
6. It seems that injection of LAKI in the dark increases the number of nocifensive behaviors (mainly wiping) as compared to saline. Is that increase significant?
7. In line 501, reported k value is different from that reported in line 768. Please provide units. In addition, what is the meaning of x? Please define Abs365n and Abs0.
8. Please add scale bar in extended data Fig. 2b.

Reviewer #3 (Remarks to the Author):

While this paper reports a novel, non-invasive and reversible tool for the study of neuronal excitability

in cutaneous primary sensory afferents, it also presents several weaknesses.

1) Primarily, the fact that LAKI acts on both TREK-1/2 and TRESK presents a major drawback. Indeed, TREK-1/2 channels are expressed in C-fibers with a bias towards peptidergic neurons, while TRESK expression is biased towards non-peptidergic C-fibers (<http://mousebrain.org/adolescent/genesearch.html>). As a result, LAKI may affect different C-fiber sub-populations depending on the behavioural response assayed. Caution should therefore be used when interpreting behavioural results.

Other concerns:

1) It is assumed here that the effect of LAKI is specific to cutaneous primary afferents, but what is known about the expression of TREK/TRESK channels in non-neuronal cell types in the skin?

2) The method is claimed to be species-independent however species differences in the pharmacology and selectivity of LAKI on mammalian K channels (ex. rodents vs. primates) were not tested.

3) While the characterization of the TRESK-specific photoblock is necessary, figure 1.d clearly indicates that LAKI has a dose-dependent effect on current inhibition. Why then a dose-response curve is not shown with TREK-1 and TREK-2?

4) In both Hargreaves and Von Frey experiments (fig.3 and 4) LAKI effects on TREK/TRESK channels were no longer observed 5 minutes after the 365 nm pulse of light. However, previous experiments (fig.1.E) showed that LAKI is bi-stable in the dark. At what point in time does this stability of the active-state cis-LAKI end in the absence of a 480 nm light pulse?

5) Lines 195-96: 'for being processed as the sensation of pain' is difficult to understand. Consider rephrasing.

Minor point:

6) Fig.4 is misleading in that it presents both mouse and *C. elegans* data. Grouping the Hargreaves and Von Frey data while creating a standalone figure for *C. elegans* may be a better option. By the way, adding a panel to explain the experimental procedure used in the *C. elegans* assay would be helpful.

We thank the reviewers for their helpful comments and suggestions. In response, we have performed additional *in vitro* and *in vivo* experiments, modified the manuscript according to these new results and otherwise clarified the text, which now fully addresses the concerns raised by the reviewers and answers the remaining questions. Altogether, we believe that the additional experiments and modifications to the text bolster our conclusions and considerably strengthen the manuscript.

Herewith we submit for your consideration a point-by-point response to each of the reviewers' concerns and we explain the changes that we have made as required.

Reviewer #1 (Remarks to the Author):

Note that I have read the entire draft including supplemental materials, my expertise is in the sensory biophysics of C. elegans. Therefore, I have no technical comments on the sections outside of this scope.

This is a well written and convincing paper on the development and testing of a photoswitchable inhibitor (LAKI) of a nociceptive-related 2-pore-domain potassium channel. In addition to tests at the chemical and neurophysiological level, the authors test LAKI at the behavioral level in standard nociceptive/pain assays using mice and C. elegans. Overall I see this work as thorough and convincing. I have to leave technical comments on the chemistry and mouse assays to other reviewers. Overall I believe this is an important contribution to "pain" research assays and will be of interest to a broad audience.

We thank the reviewer for his/her positive feedback on the broad interest and the conceptual novelty of the use of LAKI to remote control pain. We have been pleased to improve the description of our assays, to clarify how experiments have been done and to provide additional controls and characterization as suggested (see below).

Comments

Re: C. elegans assay

1) Experimental details are missing. Currently it is not possible to reproduce the data in the C. elegans experiment.

We thank the reviewer for having noted this point and we realized that information was missing to reproduce this easy test. We have now provided this information in the Material & Methods section (supplementary files), as following.

a) how are the worms illuminated with the 365 nm and 480 nm light.

The worms were recorded on a Zeiss axiovert 200 microscope under a 10X magnification. The worms were illuminated using LED lamps placed at 5 cm above the plate.

b) what are the light sources (LED or laser) and what power is used for the 480 nm light. Identifying the light sources are important because the spectrum of the LED is different depending on the manufacturer.

All the light sources we used are commercially available LED lamps. The power of the 480 nm light is $< 1\text{mW}\cdot\text{mm}^2$ ($0.62\text{ mW}\cdot\text{mm}^2$). The information regarding the origin/brand of the LEDs have been added to the text.

2) Proper controls may be missing for the pharmacological results. It is true that omega turn responses (with LAKI) are reduced when certain analgesics are applied. It is not clear if the analgesics themselves are modifying basal responses in a way to suppress omega turns. No control data is included.

We thank the reviewer to point out the absence of an important control. We followed Rev1 recommendations and tested the impact of analgesic drugs on the omega turn behavior induced by UV light in the absence of LAKI. As shown on the new Data Figure 15, all the analgesic drugs that we tested did not have a direct and significant impact on the worm endogenous sensitivity to UV light. The text was modified accordingly by adding the sentence: “This analgesic-reversed effect is related to LAKI since it is not observed in its absence”.

Extended Data Fig. 15 Analgesic does not reduce endogenous worm sensitivity to UV light.

Bar graph summarizing the relative average of Omega turns made either in presence of ML67.33 (80 μ M), Ibuprofen (100 μ M), Nefopam (100 μ M), APAP (100 μ M), AEA (100 μ M) or equivalent DMSO upon 15s illumination at 365 nm. Statistical significance was determined by QuasiPoisson GLM followed by Dunnett's post-test (non significant $p > 0.05$ versus control). Data are represented as mean \pm SEM adjusted for propagation of uncertainties. The numbers of *C. elegans* are indicated in parentheses on the graph.

3) Fig 4. The figure legend states that data are represented as "mean \pm SEM" but what about "fold" changes. Errors were propagated using standard methods?

The errors on the fold change values were obtained using the standard error propagation method. More precisely, we derived the error values ΔY from the following generic formula :

$$\Delta Y = \sqrt{\sum_i \left(\frac{\partial Y}{\partial X_i} \Delta X_i \right)^2}$$

ΔY is the error on the value $Y = f(X_i)$ that has been computed using the experimental values X_i , each of which is endowed with the corresponding ΔX_i error.

Here, for a given compound, the average fold change $Y = \frac{\mu(N_{\Omega}^{molecule})}{\mu(N_{\Omega}^{control})}$ is computed using 2 experimental values: $\mu(N_{\Omega}^{molecule})$, the average number of omega turns of *C. elegans* incubated with the compound under UV light and $\mu(N_{\Omega}^{control})$, the average number of omega turns of a *C. elegans* incubated without the compound under UV light (μ stands for the average). Therefore, the error on the average fold change, ΔY is :

$$\Delta Y = Y \sqrt{\left(\frac{SEM(N_{\Omega}^{molecule})}{\mu(N_{\Omega}^{molecule})} \right)^2 + \left(\frac{SEM(N_{\Omega}^{control})}{\mu(N_{\Omega}^{control})} \right)^2}$$

Here, SEM stands for the standard error of the mean. The SEM of $N_{\Omega}^{molecule}$ and $N_{\Omega}^{control}$ are taken as the errors of $\mu(N_{\Omega}^{molecule})$ and $\mu(N_{\Omega}^{control})$, respectively.

A sentence in the text has been added to clarify this representation.

4) L623. "...to full light." What does "full light" indicate? Full power?

"Full light" was not appropriated and we change it by "ambient light".

5) L624. when *C. elegans* "fully annealed." What is meant by "fully annealed" here?

Many thanks to point out this mistake due to the fact that the word "anneal" is a "false friend" in French. In fact, "anneal" is similar to "anneau" which means "ring" in French, thus explaining why "annealing" was improperly used. We have now modified the text by substituting it by "circling".

6) Reference [30] is incomplete.

We have modified the text accordingly.

Minor:

L615. "*E. coli*" should be italicized.

We have modified the text accordingly.

re: rate limiting step(s). I think many people would be interested in temporal limitation of the technique and perhaps it would be useful to see a magnified view (in time) of Fig 1b in order to get a better sense of the temporal constraints in de-activation and activation. A comment on the rate of photoisomerization vs functional modulation of neuronal activity would be useful.

We have indeed a temporal limitation related to the conformational change of LAKI upon a pulse of light and its binding on its target which is related to the affinity.

Accordingly, following Rev1' suggestion, we have compared the different kinetics in order to show the temporal resolution. We have now provided an additional Figure (Extended Data Fig. 5) in which the time scale was decreased to better illustrate the kinetic of block and unblock. Furthermore, we added a panel to report the blocking and unblocking time constant τ of the model used to fit TRESK current.

We observed a difference between the kinetic of *cis* and *trans* isomerization of LAKI and the functional consequences in neurons. This kinetic is related to the binding and unbinding of LAKI to the targeted channel. The association constant (k_{on}) and the dissociation constant (k_{off}) appeared to be the limiting factors.

A sentence has been added to the text:

“This light-gated block and unblock happened in average in less than 170 ms and 310 ms respectively (Extended Data Fig. 5)”.

Extended Data Fig. 5: LAKI blocking and unblocking kinetics on TRESK channel.

(a) Representative current recording elicited at 0 mV from HEK293T cells expressing TRESK in presence of LAKI (5 μ M), upon alternating illumination at 480 nm (blue) and 365 nm (magenta). (b) Bar graph summarizing the LAKI blocking (365 nm) and unblocking (480 nm) time constants tau on TRESK channel at 0 mV. Data are represented as mean \pm SEM. The numbers of tested cells are indicated in parentheses on the graph.

Reviewer #2 (Remarks to the Author):

In this study, Landra-Willm and colleagues design a photoswitchable inhibitor of K2P TREK and TRESK channels, named LAKI, and show that it reversibly controls animal behavior in response to different painful stimuli. The authors synthesized LAKI by an “azologization”-based approach, and found that the compound is inactive in vitro in the dark, but active at 365 nm light by inhibiting selected K2P channels (TREK1, TREK2 and TRESK). They next used LAKI in vivo, in both mice and C. elegans, to show reversible optical control of pain or escape behavior, thus demonstrating the usefulness of their tool in different species. Finally, they provide in vivo evidence by extracellular recordings that LAKI regulates C-fiber excitability in mice. The main advance of LAKI is its inactive behavior in the dark, as compared to previously reported photochromic ligands which lock ion channels in ambient light. Overall, I find the study well executed and technically sound.

We first would like to thank the reviewer for his/her positive feedback on our work.

However, my main concern is the lack of evidence that TREK/TESK are effectively targeted in vivo by LAKI. Although the authors provide pharmacological evidence with ML67.33, it is not sufficient to support the claim that local acute channel closing of TREK/TRESK in mice is sufficient to induce pain behavior. In addition, the following comments need to be addressed.

Regarding the specificity of LAKI, we agree that determining the specificity based only on ML67-33 is not strong enough. However, we assessed LAKI-induced photocurrent on isolated nociceptors from native Trigeminal Ganglia (TG) from either Wild-Type mice or mice in which both *Trek1* and *Trek2* were genetically invalidated (double KO mice). In addition, in a third condition, we overexpressed in *Trek1*^{-/-}/*Trek2*^{-/-} TG neurons TRESK-MT1, a dominant negative form to silence TRESK (see Royal et al., 2019). As shown on Fig. 2, *Trek1* and 2 genetic invalidation suppressed 77% of the light gated current and the expression of the TRESK-MT1 suppressed an additional 48% of the remaining current. This permits to conclude that the current gated by light in these neurons is almost exclusively carried by TREK1, TREK2 and TRESK.

In vivo, we used the ML67-33 compound to demonstrate the specificity regarding TREK channels. Nevertheless, we agree with Rev1 that it was not sufficient to assess that LAKI is acting through TREK channels *in vivo*.

To further demonstrate the implication of TREK channels, to fully address Rev2’s concerns, **we have now added behavioral experiments carried out using the *Trek1/Trek2* double KO animals and we found that *Trek* genetic invalidation is sufficient to suppress allodynia induced by LAKI *cis* isomerization (see below).**

Together, these data now demonstrate that the channels which are controlled by LAKI, in the context of small diameter sensory neurons and pain, are TREK1, TREK2 and TRESK.

Comments

1. The authors provide strong evidence that TREK1 and TREK2 are targeted by LAKI *in vitro*, as they report 80% decrease of light-blocked current in $T1^{-}/T2^{-}$ TG neurons. However, they do not really demonstrate this is the case *in vivo* (e.g. in mice), although they used ML67.33 to pharmacologically prevent light-induced thermal hypersensitivity in wild-type mice. To further support the contribution of these channels, the authors should show the absence of *in vivo* effects of LAKI in $T1^{-}/T2^{-}$ mice.

As explained above, we have conducted Von Frey experiments on *Trek1* and *Trek2* double KO animals and tested the effects of light on LAKI-injected foot pad.

As shown in the new Extended Data Fig. 13, *Trek1* and *Trek2* genetic invalidation was sufficient to suppress the allodynia induced by LAKI *cis* isomerization (see below).

Extended Data Fig. 13 LAKI controls mechanical allodynia through the inhibition of TREK channels. (a) Bar graph summarizing the relative paw withdrawal threshold of either WT or *Trek1*^{-/-}/*Trek2*^{-/-} double KO mice injected with LAKI (100 μ M, 15 μ L) in the dark or after 20s illumination at 365 nm (magenta). Statistical significance was determined by Mixed-effects model with repeated measures followed by Holm-Sidak's post-test (***) $p < 0.001$ before and after illumination at 365 nm). (b) Graph summarizing the average of the paw withdrawal threshold of WT and *Trek1*^{-/-}/*Trek2*^{-/-} double KO mice injected with LAKI (100 μ M, 15 μ L) relatively to mice before illumination at 365 nm, in the dark, after 20s illumination at 365 nm

(magenta) and after 4 min in dark. Statistical significance was determined by Mixed-effects model with repeated measures followed by Holm-Sidak's post-test (### $p < 0.001$ between WT and *Trek1^{-/-}/Trek2^{-/-}* mice). Data are represented as mean \pm SEM. The numbers of mice are indicated in parentheses on the graph.

The text was modified accordingly with an additional sentence:

“This UV light-induced allodynia was prevented by *Trek1/Trek2* genetic invalidation ($P > 0.99$), further supporting TREK channel involvement.”

2. The chemical structure of LAKI only differs from that of ML365 by a relatively small chemical substitution (amide versus diazene). However, the pharmacological profile of LAKI seems completely different from that of ML365 in respect to TASK1 and TASK3 channels (no inhibition induced by LAKI for TASK1 and TASK3, whereas ML365 is a full inhibitor of TASK1 and TASK3). How is it possible? Please comment.

The initial disclosure of ML365 did not thoroughly characterize its pharmacology, even among K_{2P} channels, and when we further evaluated its activity, we found that ML365 also inhibits TREK1/2 and TRESK, but not TRAAK (Extended Data Figure 1), so there is overlap in terms of which channels ML365 and LAKI can inhibit. Generally, using an azo group as an isostere for an amide works, because the amide and trans-azo dihedral angles are similar when molecules containing these moieties bind to their targets (see e.g. Trauner's 2019 mapping of the azo isostere space DOI: 10.1021/acscentsci.8b00881). However, the overlap is not perfect, and there are other, less common dihedral angles that azo groups can adopt that are very rarely seen with biaryl amides. Relatively small functional group changes can have large effects on the binding pose of a ligand; switching from a biaryl amide to a biaryl sulphonamide significantly changes the dihedral angle, so much so that biarylsulphonamides are considered reasonable isosteres for *cis*-azo groups (Kobauri et al., 2021). The amides are slightly bigger than the diazen and the angle of the rings regarding each other's is modified. Replacement of ester by amide in ML365 reduces the proportion of gauche conformer about the C--C bond. Additionally, ligand binding to its receptor depends on some degree of flexibility in the --CO--O--bond, though the hydration of the bond may also be important. Therefore, this small modification may explain the difference in the selectivity of LAKI compared to ML365.

3. Photochemical data show that the half-life of cis-LAKI at 21.5°C was 0.7 hours (i.e. 42 min). However, in vivo data obtained in mice (under both thermal and Von Frey assays) rather suggest a complete relaxation process in less than 5 minutes. How can the authors explain this difference? Is it related to temperature difference (21.7°C in vitro versus 37°C in vivo)? It would be good to measure half-life of LAKI at 37°C.

We thank the Rev2 for this critical point and we are happy to deeply explain why we observed a complete physiological reversion after 5 mins.

As mentioned by Rev2, the temperature increase is known to speed up the spontaneous relaxation. We also believed at first that the temperature was the primary explanation for this relatively fast loss of LAKI effect *in vivo* but it is not. In our case, this temperature increase leading to a reduction of half-life at 37°C plays only a negligible role that cannot explain this important modulation. In fact, when we used LAKI at 37°C *in vitro*, we observed no decrease of LAKI bistability assessed on TRESK channel current (Extended Data Fig. 8d).

This observed *in vivo* kinetic is rather due to a decreased concentration of photoisomerized *cis*-LAKI at the level of the nerve endings present in the epidermis.

Indeed, it is worth noting that in the *in vivo* experiments as well as in the *in vitro* experiment carried out on TRESK channel, the readout of the experiment is not directly the concentration of *cis*-LAKI but the effect of *cis*-LAKI on channels (i.e physiological response *in vivo* and TRESK current photoblock *in vitro*). The latter is directly related to the affinity characteristics of *cis*-LAKI on TRESK as shown on the concentration-response curve (Extended Data Fig. 9), the inhibitory effect of *cis*-LAKI on TRESK is observed within a one-log scale range, and can therefore be seen as a quasi-threshold relationship around the IC₅₀. Therefore, to understand the effect of LAKI on TRESK and on the physiological response, it is relevant to compare the *cis*-LAKI concentration to the IC₅₀ threshold.

This discrepancy between the photochemical and *in vivo* data can be explained as the initial concentration of *cis*-LAKI is low in the *in vivo* condition. Indeed, even though the relaxation kinetics, and therefore the **relative** *cis*-LAKI concentration, does not depend on the initial concentration of *cis*-LAKI, the **effective** *cis*-LAKI concentration over time **depends on its initial concentration** (see diagram). Therefore, the smaller the initial *cis*-LAKI concentration

is, the faster it reaches the physiological relevant concentration IC_{50} threshold and the faster the physiological effect is lost.

Diagram R1: Diagram explaining the impact of different initial *cis*-LAKI concentrations on the time at which the *cis*-LAKI concentration reaches the IC_{50} threshold through relaxation. Below the IC_{50} threshold, no *in vivo* physiological effect can be seen. Starting from a lower concentration $C_{i,1}$, the inhibitory effect of LAKI is lost before it is the case in the situation in which the initial *cis*-LAKI concentration $C_{i,2}$ is higher.

To have such reversion of the effect within short period of time, as explained above, the LAKI concentration should be within the IC_{50} range. As demonstrated below, we have evaluated that the *in vivo* concentration of *cis*-LAKI turns out to be close to the IC_{50} threshold.

This decreased concentration of *cis*-LAKI is the result of a combination of two factors which are (i) the concentration of LAKI and (ii) the absorption of light by the skin.

(i) The LAKI concentration: First, the concentration of LAKI in mouse paw is less than the injected concentration of $100 \mu\text{M}$ because of LAKI dilution in the volume of the paw. Considering a standard volume of $120 \mu\text{L}$ for mouse paw, injection of $15 \mu\text{L}$ of LAKI $100 \mu\text{M}$ will result to a concentration $\sim 10\mu\text{M}$ which already reduces the pool of LAKI available for photoconversion at nerve endings.

(ii) The intensity of light: Second, in the skin, the intensity of UV-A light strongly decreases with the distance of penetration. Considering mouse hind paw skin thickness (stratum corneum plus epidermis) of approximately $80 \mu\text{m}$ (Wai et al., 2021), as shown on figure R1c (Finlayson et al., 2022), only 25% of light reaches the epidermis in which the nerve endings are located.

[redacted]

Figure R1 : Normalized fluence rate as a function of depth into the skin for incident light at angles of incidence of 0, 45 and 65° from the normal to the skin surface (From Finlayson et al., 2021). The results are plotted for wavelengths of (a) 222 nm, (b) 300 nm, (c) 350 nm, (d) 400 nm, (e) 630 nm and (f) 800 nm. Data are taken from a central column of the outputted fluence rate grid. For all wavelengths, the most oblique angles result in a higher fluence rate at the top of the skin but an overall smaller penetration depth.

Therefore, based on these parameters, only ~ 3 μM of LAKI would be photoisomerized and be active. At this concentration, which is close to the IC_{50} , the time necessary to reach the IC_{50} threshold, for which the inhibitory effect of *cis*-LAKI is negligible, is largely reduced, which explains the *in vivo* loss of effect of *cis*-LAKI within 5 minutes.

In order to test this hypothesis, we assessed the bistability of LAKI effect on TRESK current and modified 2 parameters which are the concentration and the UV light intensity.

Consistently with Figure 1e, LAKI 10 μM effect on TRESK current is bistable upon a pulse of light at 365 nm with an intensity of $4.32 \text{ mW}\cdot\text{cm}^{-2}$. However, we observed a faster LAKI loss of effect with a time constant τ of approximately 3 minutes when the UV light intensity was decreased to $1.05 \text{ mW}\cdot\text{cm}^{-2}$. (Extended Data Fig. 8a,b). Moreover, we confirmed that this faster LAKI loss of effect was indeed due to the quantity of photoisomerized LAKI since the bistability of LAKI effect was rescued when LAKI concentration was increased to 100 μM (Extended data Fig. 8c).

Extended Data Fig. 8: LAKI functional bistability is dependent of UV light intensity and concentration. (a) Representative whole-cell current recordings elicited at -60 mV from HEK293T expressing TRESK in the presence of LAKI (10 μ M) upon illumination at 480 nm (blue) and 365 nm (magenta) or in the dark. Dark and grey traces represent cell illuminated respectively with 4.32 and 1.05 $\text{mW}\cdot\text{cm}^{-2}$ UV light. (b) Bar graph showing the time constant τ of LAKI relaxation on TRESK channel after 1.05 $\text{mW}\cdot\text{cm}^{-2}$ illumination at 365 nm. (c) Representative whole-cell current recordings elicited at -60 mV from HEK293T expressing TRESK in the presence of LAKI (100 μ M) upon illumination at 480 nm (blue) and 365 nm (magenta) or in the dark. (d) Representative whole-cell current recordings elicited at -60 mV from HEK293T expressing TRESK perfused with *cis*-LAKI (10 μ M) maintained at 37°C. Black arrow represents *cis*-LAKI perfusion. Data are represented as mean \pm SEM. The numbers of tested cells are indicated in parentheses on the graph.

A paragraph has been added: “This long bistability relies on the effective concentration of *cis*-LAKI. Indeed, we observed a decrease of the block of TRESK channel over the time with a tau of ~3 min when UV light intensity was decreased to mimic *in vivo* UV light illumination (Extended Data Fig. 8a, 8b).”

4. In line 123, can the authors provide the actual inhibition level in native nociceptors? Is that inhibition level comparable to that obtained in the recombinant system (about 80%)? In Fig. 2a, please add the zero-current level.

It is not comparable. Indeed, in HEK cells, the majority of the channels present at the surface of the cell are close to be exclusively the transfected (overexpressed) ones. In this recombinant system, the current observed is mostly carried by the transfected channel with a negligible part related to the endogenous ones. In the case of neurons, we are working on native cells which exhibit all kind of conductances and in which TREK and TRESK channels are expressed at their physiological level. Therefore, in neurons, endogenous currents, in addition to $I_{\text{TREK-TRESK}}$, are not negligible. As a consequence, the neuron cannot be used to determine the efficacy of LAKI on TREK and TRESK channels. However, it can be used to determine what is the relative contribution of $I_{\text{TREK-TRESK}}$ to the total observed current for a given potential. We found that ~26% of the total current observed at -60 mV is blocked by light, indicating a contribution of TREK1/2 and TRESK of, at least, 26% of the neuronal leak current.

The traces showing the current at different potentials in Fig. 2a are illustrating the variation of the light-blocked current depending on the membrane potential. The IV curve on Panel b represents the ΔI ($I_{480\text{nm}} - I_{365\text{nm}}$) as a function of different V_m and shows the reversion of the current at the expected potential for a potassium current (i.e -80 mV). Representative traces at the different potentials with the baseline showing the zero current would not be appropriate since it would squeeze the light-gated current part, making hard to see it for the lower potentials.

To avoid confusion and to show the relative contribution of TREK and TRESK to the total leak current, we provided a supplementary Figure representing the light-blocked current at -40 mV with the zero current baseline and the associated LAKI-induced current inhibition (Extended Data Fig. 10). For the bar graph, -40 mV and -60 mV were chosen to avoid the window current of voltage-gated channels and therefore to decrease their contribution.

A sentence has been added: “which represents ~ 26% of the neuronal leak current observed at -60 mV (Extended Data Fig. 10).”.

Extended Data Fig. 10: TREK channels contribution to neuronal leak current.

(a) Whole-cell current recording elicited at -40 mV from WT TG neurons in presence of LAKI (10 μ M) upon alternating illumination at 480 nm (blue) and 365 nm (magenta). (b) Bar graph summarizing the contribution of TREK1/2 and TRESK to WT TG neuron leak current at -40 and -60 mV. Data are represented as mean \pm SEM. The numbers of tested neurons are indicated in parentheses on the graph.

5. In Figs. 1f, 2a and 2c, why is there an “off-inhibition” effect under the first irradiation at 480 nm? Was the cell previously irradiated at 365 nm before recording?

Rev2 is right and we thank him/her for this remark, there is indeed an “off-inhibition” due to our graphical representation which may be misleading. This is due to the UV exposition preceding the 480 nm illumination. We recorded currents in gap-free mode with continuous alternating illumination and after gating 3-4 cycles, we increased the Vm by 20 mV. In order to avoid this confusion, we have now modified the figure accordingly by adding a small line colored in magenta.

New Figure 2 :

Fig. 2: LAKI specifically photoblocks TREK1, TREK2 and TRESK in native small sensory neurons. (a) Whole-cell current recordings elicited at different holding potentials from wild-type (WT) TG neurons in presence of LAKI (10 μ M) upon alternating illumination at 480 nm (blue) and 365 nm (magenta). (b) IV relationship of the photocurrent density induced by alternating illumination ($I_{480\text{ nm}} - I_{365\text{ nm}}$) in WT TG neurons in the presence of LAKI (10 μ M). (c) Whole-cell current recordings elicited at 0 mV from WT TG neurons, $Trek1^{-/-}/Trek2^{-/-}$ TG neurons and $Trek1^{-/-}/Trek2^{-/-}$ TG neurons overexpressing TRESK-MT1 in presence of LAKI (10 μ M) upon alternating illumination at 480 nm (blue) and 365 nm (magenta). (d) IV relationship of the photocurrent density ($I_{480\text{ nm}} - I_{365\text{ nm}}$) in WT TG neurons, $Trek1^{-/-}/Trek2^{-/-}$ TG neurons and $Trek1^{-/-}/Trek2^{-/-}$ TG neurons overexpressing TRESK-MT1 in presence of LAKI (10 μ M). Statistical significance was determined by Mixed-effects model with repeated measures followed by Holm-Sidak's post-test (***) $p < 0.001$ compared WT TG with $Trek1^{-/-}/Trek2^{-/-}$ TG and $Trek1^{-/-}/Trek2^{-/-}$ TG + TRESK-MT1; ### $p < 0.001$, # $p < 0.05$ compared $Trek1^{-/-}/Trek2^{-/-}$ TG with $Trek1^{-/-}/Trek2^{-/-}$ TG + TRESK-MT1). (e) Representative voltage trace showing the

firing modulation of WT TG neurons in presence of LAKI (10 μM) upon alternating illumination at 480 nm (blue) and 365 nm (magenta). (f) Graph summarizing the average of the firing rate of WT TG neurons. Statistical significance was determined by paired t-test (** $p < 0.01$). Data are represented as mean \pm SEM. The numbers of tested neurons are indicated in parentheses on the graph.

6. It seems that injection of LAKI in the dark increases the number of nocifensive behaviors (mainly wiping) as compared to saline. Is that increase significant?

We also noticed this small increase. We have used a Poisson GLM followed by Bonferroni's post-test and found it is not significant ($P > 0.26$).

7. In line 501, reported k value is different from that reported in line 768. Please provide units. In addition, what is the meaning of x ? Please define Abs_{365n} and Abs_0 .

We first thank the reviewer for having noticed this error on the k value which equals 0.99 h^{-1} . x was used as multiplicative sign, we replaced it to avoid confusion.

Here, we provide more informations about the equation used to determine LAKI half-life :

To measure the thermal relaxation (*cis* to *trans* isomerization) of a solution of LAKI (5 μM in HEPES solution containing 0.1% DMSO), the solution was irradiated with $\text{UV}_{365\text{nm}}$ light to achieve the PSS_{365} . LAKI was allowed to thermally relax at 21.5°C and was monitored at 330 nm for 20 hours. Absorbance was plotted vs time and thermal half-life was calculated using $t_{1/2} = 0.693/k$. The Solver add-on in Microsoft Excel was used to find a line fit using the GRG nonlinear engine using the equation $Abs(t) = Abs_{S_{BI}} + (Abs_{365\text{nm}} - Abs_{S_{BI}}) * \exp(-kt)$, where k is the rate constant (h^{-1}), t is time (h), $Abs_{S_{BI}}$ is absorbance before illumination, and $Abs_{365\text{nm}}$ is absorbance at PSS_{365} . The variables $Abs_{S_{BI}}$, $Abs_{365\text{nm}}$ and k were used as free parameters in the fit.

The text was modified accordingly.

8. Please add scale bar in extended data Fig. 2b.

The scale has been now added to the new Figure 2 (see above).

Extended Data Fig. 2: Photochemical characterization of LAKI.

(a) UV-Visible absorbance spectra of LAKI (5 μM in HEPES solution containing 0.1% DMSO) before and after photoirradiation (blue 460 nm light and UV 365 nm light). Inset shows the rate of switching between blue 460 nm light photostationary state and UV 365 nm light photostationary state. (b) $^1\text{H-NMR}$ spectra of LAKI collected in CD_3CN showing the photostationary state. (c) Thermal half-life of LAKI (5 μM in HEPES solution containing 0.1% DMSO) observed at 330 nm at 21.5°C to yield a half-life of 0.7 hours ($k = 0.99 \text{ h}^{-1}$). (d) Photocycling of LAKI observed at 360 nm with alternate irradiation of blue 460 nm light and UV 365 nm light for 5 min each.

Reviewer 3

While this paper reports a novel, non-invasive and reversible tool for the study of neuronal excitability in cutaneous primary sensory afferents, it also presents several weaknesses.

We thank the reviewer for his/her positive feedback on our manuscript.

1) Primarily, the fact that LAKI acts on both TREK-1/2 and TRESK presents a major drawback. Indeed, TREK-1/2 channels are expressed in C-fibers with a bias towards peptidergic neurons, while TRESK expression is biased towards non-peptidergic C-fibers (<http://mousebrain.org/adolescent/genesearch.html>). As a result, LAKI may affect different C-fiber subpopulations depending on the behavioural response assayed. Caution should therefore be used when interpreting behavioural results.

In the mouse brain database, TRESK and TREK1 do not seem to be expressed in the same neurons and TRESK is absent from the ones which are negative for IB4-. This is not fitting to experiments previously done in the lab in which we found that both TREK1/2 are expressed in peptidergic and non-peptidergic neurons (Avalos Prado et al., 2021). Others have shown functional expression of TRESK in both subpopulations (Guo et al., 2019) while we demonstrated the functional coexpression of TREK1/2 and TRESK (Royal et al., 2019). It was also shown by Weir and collaborators that TRESK is expressed in both IB4+ and IB4- neurons with a preferential expression of TRESK in the IB4+ ones, however, that is also observed for TREK2 (Chang et al., 2021). Furthermore, others reported a uniform representation of TRESK in the different neuronal subpopulations (Dobler et al., 2007 ; Yoo et al., 2009). This may be a question of threshold of expression.

As shown on the Figure R2, immunolabelling of TREK1, TREK2 and TRESK in TG neuron primary culture confirms that the 3 proteins are co-expressed in the same neurons. In addition, we performed single cell RT-PCR on 4 different TG neurons and found that *Trek1*, *Trek2* and *Tresk* are co-expressed at relatively similar levels in the same cells.

[redacted]

Figure R2 TREK1, TREK2 and TRESK are co-expressed in TG neurons (from Royal et al., 2019). Immunodetection of TREK1, TREK2 and TRESK. Inset, bar graph representing the average relative mRNA expression of TREK1, TREK2, TRESK obtained from 4 single cell semi-quantitative RT-PCRs.

We have now added qPCR measurements to determine the relative expression levels of the different channels in Trigeminal Ganglia from WT mice. The relative expression level is represented below (Figure R3). As seen with single cell measurements, we found again that all three channels are expressed to similar levels.

Figure R3: Expression level of TRESK, TREK1 and TREK2 mRNA levels relative to TRESK.

Other concerns:

1) It is assumed here that the effect of LAKI is specific to cutaneous primary afferents, but what is known about the expression of TREK/TRESK channels in non-neuronal cell types in the skin?

There are some indications based on single cell RNA sequencing that TREK channels are expressed in keratinocytes (epidermis) and fibroblasts (dermis) (Solé-Boldo et al., 2020) and TREK channels are also expressed in HaCaT cells which is an immortalized keratinocyte cell

line (Kang et al., 2007). Keratinocytes can release several compounds, such as calcitonin gene-related peptide (CGRP), adenosine triphosphate (ATP), acetylcholine, glutamate, various growth factors, cytokines, chemokines, and many other autacoids that play a role in pain perception and inflammation.

Therefore, we cannot exclude a role of the light at this level and a role of TREK channels in keratinocyte function in pain.

2) The method is claimed to be species-independent however species differences in the pharmacology and selectivity of LAKI on mammalian K channels (ex. rodents vs. primates) were not tested.

To address the concern of Rev3, we have now compared the efficacy of LAKI on the currents elicited by both mouse and human TRESK and TRAAK orthologues.

An additional Extended Figure 4 has been provided and a sentence in the text:

“LAKI photoblock is conserved among species since TRESK and TRAAK channel inhibitions are similar in mouse and human (Extended Data Fig. 4)”

Extended Data Fig. 4 LAKI is similarly efficient to photoblock TRESK and TRAAK channels from different species

(a) Normalized whole-cell current recording elicited at -60 mV from HEK293T cells expressing either mouse or human TRESK in the presence of LAKI (5 μ M) upon alternating illumination at 480 nm (blue) and 365 nm (magenta). (b) Bar graph summarizing the current inhibition (%) of mouse and human TRESK at -60 mV. (c) Normalized whole-cell current recording elicited at -60 mV from HEK293T cells expressing either mouse or human TRAAK in the presence of LAKI (5 μ M) upon alternating illumination at 480 nm (blue) and 365 nm (magenta). (d) Bar graph summarizing the current inhibition (%) of mouse and human TRAAK at -60 mV. Statistical significance was determined by QuasiBinomial GLM (non significant $p > 0.05$). Data are represented as mean \pm SEM. The numbers of tested cells are indicated in parentheses on the graph.

3) While the characterization of the TRESK-specific photoblock is necessary, figure 1.d clearly indicates that LAKI has a dose-dependent effect on current inhibition. Why then a dose-response curve is not shown with TREK-1 and TREK-2?

To respond to this point, we have conducted new experiments to obtain the concentration-response of LAKI on TREK1 and TREK2 channels. The results obtained are now shown in the new Extended Figure 9.

Extended Data Fig. 9: LAKI concentration-response curves on TREK1, TREK2 and TRESK channel.

Normalized concentration-response curve of LAKI on whole-cell current elicited at -60 mV from HEK293T cells expressing either (a) TREK1, (b) TREK2 or (c) TRESK. Data points were fitted using a four-parameter logistic curve from $n = 3-7$ independent cells. Data are represented as mean \pm SEM.

4) In both Hargreaves and Von Frey experiments (fig.3 and 4) LAKI effects on TREK/TRESK channels were no longer observed 5 minutes after the 365 nm pulse of light. However, previous experiments (fig.1.E) showed that LAKI is bi-stable in the dark. At what point in time does this stability of the active-state cis-LAKI end in the absence of a 480 nm light pulse?

We thank the Rev3 for this important point which was also pointed out by Rev2. In order to respond to Rev3 we have conducted *in vitro* experiments and tested the impact of the temperature as well as the LAKI concentration on the functional bistability.

As this important point was also raised by rev2, you will find below a copy of the response made above :

As mentioned by Rev2 the temperature increase is known to speed up the spontaneous relaxation. We also believed at first that the temperature was the primary explanation for this relatively fast loss of LAKI effect *in vivo* but it is not. In our case, this temperature increase leading to a reduction of half-life at 37°C plays only a negligible role that cannot explain this important modulation. In fact, when we used LAKI at 37°C *in vitro*, we observed no decrease of LAKI bistability assessed on TRESK channel current (Extended Data Fig. 8d).

This observed *in vivo* kinetic is rather due to a decreased concentration of photoisomerized *cis*-LAKI at the level of the nerve endings present in the epidermis.

Indeed, it is worth noting that in the *in vivo* experiments as well as in the *in vitro* experiment carried out on TRESK channel, the readout of the experiment is not directly the concentration of *cis*-LAKI but the effect of *cis*-LAKI on channels (i.e physiological response *in vivo* and TRESK current photoblock *in vitro*). The latter is directly related to the affinity characteristics of *cis*-LAKI on TRESK: as shown on the concentration-response curve (Extended Data Fig. 9), the inhibitory effect of *cis*-LAKI on TRESK is observed within a one-log scale range, and can therefore be seen as a quasi-threshold relationship around the IC₅₀. Therefore, to understand the effect of LAKI on TRESK and on the physiological response, it is relevant to compare the *cis*-LAKI concentration to the IC₅₀ threshold.

This discrepancy between the photochemical and *in vivo* data can be explained as the initial concentration of *cis*-LAKI is low in the *in vivo* condition. Indeed, even though the relaxation kinetics, and therefore the **relative** *cis*-LAKI concentration, does not depend on the initial concentration of *cis*-LAKI, the **effective** *cis*-LAKI concentration over time depends on its initial concentration (see diagram). Therefore, the smaller the initial *cis*-LAKI concentration is, the faster it reaches the physiological relevant concentration IC₅₀ threshold and faster the physiological effect is lost.

Diagram R1: Diagram explaining the impact of different initial *cis*-LAKI concentrations on the time at which the *cis*-LAKI concentration reaches the IC_{50} threshold through relaxation. Below the IC_{50} threshold, no *in vivo* physiological effect can be seen. Starting from a lower concentration $C_{i,1}$, the inhibitory effect of LAKI is lost before it is the case in the situation where the initial *cis*-LAKI concentration $C_{i,2}$ is higher.

To have such reversion within the effect, as explained above, the LAKI concentration should be within the IC_{50} range. As demonstrated below, we have evaluated that the *in vivo* concentration of *cis*-LAKI turns out to be close to the IC_{50} threshold.

This decreased concentration of *cis*-LAKI is the result of a combination of two factors which are (i) the concentration of LAKI and (ii) the absorption of light by the skin.

(i) **The LAKI concentration:** First, the concentration of LAKI in mice paw is less than the injected concentration of 100 μM because of LAKI dilution in the volume of the paw. Considering a standard volume of 120 μL for mouse paw, injection of 15 μL of LAKI 100 μM will result to a concentration ~ 10 which already reduces the pool of LAKI available for photoconversion at nerve endings.

(ii) **The intensity of light:** Second, in the skin, the intensity of UV-A light strongly decreases with the distance of penetration. Considering mouse hind paw skin thickness (stratum corneum plus epidermis) of approximately 80 μm (Wai et al., 2021), as shown on figure R1c (Finlayson et al., 2022), only 25% of light reaches the epidermis in which the nerve endings are located.

[redacted]

Figure R1 : Normalized fluence rate as a function of depth into the skin for incident light at angles of incidence of 0, 45 and 65° from the normal to the skin surface (From Finlayson et al., 2021). The results are plotted for

wavelengths of (a) 222 nm, (b) 300 nm, (c) 350 nm, (d) 400 nm, (e) 630 nm and (f) 800 nm. Data are taken from a central column of the outputted fluence rate grid. For all wavelengths, the most oblique angles result in a higher fluence rate at the top of the skin but an overall smaller penetration depth.

Therefore, based on these parameters, only $\sim 3 \mu\text{M}$ of LAKI would be photoisomerized and be active. At this concentration, which is close to the IC_{50} , the time necessary to reach the IC_{50} threshold, for which the inhibitory effect of *cis*-LAKI is negligible, is largely reduced which explains the *in vivo* loss of effect of *cis*-LAKI by 5 minutes.

In order to test this hypothesis, we assessed the bistability of LAKI's effect on TRESK current and modified 2 parameters which are the concentration and the UV light intensity.

Consistently with Figure 1e, LAKI $10 \mu\text{M}$ effect on TRESK current is bistable upon a pulse of light at 365 nm with an intensity of $4.32 \text{ mW}\cdot\text{cm}^{-2}$. However, we observed a faster LAKI loss of effect with a time constant τ of approximately 3 minutes when the UV light intensity was decreased to $1.05 \text{ mW}\cdot\text{cm}^{-2}$. (Extended Data Fig. 8a,b). Moreover, we confirmed that this faster LAKI loss of effect was indeed due to the quantity of photoisomerized LAKI since the bistability of LAKI effect was rescued when LAKI concentration was increased to $100 \mu\text{M}$ (Extended data Fig. 8c).

Extended Data Fig. 8: LAKI bistability is dependent of UV light intensity and concentration. (a) Representative whole-cell current recordings elicited at -60 mV from HEK293T expressing TRESK in the presence of LAKI (10 μ M) upon illumination at 480 nm (blue) and 365 nm (magenta) or in the dark. Dark and grey traces represent cell illuminated respectively with 4.32 and 1.05 mW.cm⁻² UV light. (b) Bar graph showing the time constant τ of LAKI relaxation on TRESK channel after 1.05 mW.cm⁻² illumination at 365 nm. (c) Representative whole-cell current recordings elicited at -60 mV from HEK293T expressing TRESK in the presence of LAKI (100 μ M) upon illumination at 480 nm (blue) and 365 nm (magenta) or in the dark. (d) Representative whole-cell current recordings elicited at -60 mV from HEK293T expressing TRESK perfused with *cis*-LAKI (10 μ M) maintained at 37°C. Black arrow represents *cis*-LAKI perfusion. Data are represented as mean \pm SEM. The numbers of tested cells are indicated in parentheses on the graph.

A paragraph has been added: “This long bistability relies on the effective concentration of *cis*-LAKI. Indeed, we observed a decrease of the block of TRESK channel over the time with a tau of ~3 min when UV light intensity was decreased to mimic *in vivo* UV light illumination (Extended Data Fig. 8a, 8b).

5) Lines 195-96: 'for being processed as the sensation of pain' is difficult to understand. Consider rephrasing.

This was changed for :

“to be integrated and felt as pain sensation.”

Minor point:

6) Fig.4 is misleading in that it presents both mouse and *C. elegans* data. Grouping the Hargreaves and Von Frey data while creating a standalone figure for *C. elegans* may be a better option. By the way, adding a panel to explain the experimental procedure used in the *C. elegans* assay would be helpful.

We have split the Figure 4 in two figures and now the experiments determining the mechanical pain are in the new Fig. 4 whereas the experiments on *C. elegans* are illustrated in the new fig. 5, with a scheme explaining the experimental design.

Fig. 4: LAKI controls mechanical allodynia in mice by sensitization of nociceptor excitability. (a) Bar graph summarizing the paw withdrawal threshold of mice injected either with saline solution or with LAKI (100 μ M, 15 μ L) in the dark or after 20s illumination at 365 nm (magenta). Statistical significance was determined by Mixed-effects model with repeated

measures followed by Holm-Sidak's post-test (*** $p < 0.001$). (b) Graph summarizing the average of the paw withdrawal threshold of mice injected with LAKI (100 μM , 15 μL) relatively to mice injected with saline solution, in the dark or after 20s illumination at 365 nm (magenta). Statistical significance was determined by Mixed-effects model with repeated measures followed by Holm-Sidak's post-test (*** $p < 0.001$). (c) Representative spinal dorsal horn neuron responses evoked by electrical stimulation (4 mA) before and after 20s illumination at 365 nm (magenta) in mice injected with LAKI (100 μM , 15 μL). (d) Raster plot and superimposed post-stimulus histogram of dorsal horn neurons before (black) and after illumination at 365 nm (magenta) in mice injected with LAKI (100 μM , 15 μL). Data are represented as mean \pm SEM. The numbers of mice are indicated in parentheses on the graph.

Fig. 5: LAKI enables optical control of conserved pain pathways in *C. elegans*.

(a) Schematic of the experimental behavioral assay. The black arrow represents the incubation of *C. elegans* in saline or 100 μ M LAKI solution. Magenta arrow represents the beginning of 15s application of light at 365 nm. The blue arrow represents the beginning of 15s application of light at 480 nm. (b) Representative observed behavior in the dark (i), then during 365 nm light application (ii) and during 480 nm light illumination (iii). (c) Graph summarizing the fold increase generation of Omega turns under 365 nm or 480 nm illumination compared to dark, in the presence of LAKI (100 μ M) or with saline solution. Statistical significance was determined by QuasiPoisson GLM followed by Bonferroni's post-test (***) $p < 0.001$). (d) Bar graph summarizing the relative average of Omega turns made either in presence of LAKI (100 μ M), LAKI (100 μ M) plus ML67.33 (80 μ M), LAKI (100 μ M) plus Ibuprofen (100 μ M), LAKI (100 μ M) plus Nefopam (100 μ M), LAKI (100 μ M) plus APAP (100 μ M), LAKI (100 μ M) plus AEA (100 μ M) or equivalent DMSO upon 15s illumination at 365 nm. Statistical significance was determined by QuasiPoisson GLM followed by Dunnett's post-test (* $p < 0.05$, *** $p < 0.001$ versus control ; # $p < 0.05$, ### $p < 0.001$ versus LAKI). Data are represented as mean \pm

SEM adjusted for propagation of uncertainties. The numbers of *C. elegans* are indicated in parentheses on the graph.

REVIEWERS' COMMENTS

Reviewer #1 (Remarks to the Author):

The revision answers all of my questions and comments with the original draft. I have no more concerns and feel the paper is suitable for publication.

Reviewer #2 (Remarks to the Author):

The authors have satisfactorily addressed all my points. They have done an excellent job and I am satisfied with the revised manuscript. I have no further comment to make. I recommend accepting the manuscript for publication.

Reviewer #3 (Remarks to the Author):

The authors have addressed all of the major concerns expressed by the reviewers. The manuscript has been significantly improved.